# Optimal Best-arm Identification in Linear Bandits

**Yassir Jedra**
KTH
Stockholm, Sweden
jedra@kth.se

**Alexandre Proutiere**
KTH
Stockholm, Sweden
alepro@kth.se

## Abstract

We study the problem of best-arm identification with fixed confidence in stochastic linear bandits. The objective is to identify the best arm with a given level of certainty while minimizing the sampling budget. We devise a simple algorithm whose sampling complexity matches known instance-specific lower bounds, asymptotically almost surely and in expectation. The algorithm relies on an arm sampling rule that tracks an optimal proportion of arm draws, and that remarkably can be updated as rarely as we wish, without compromising its theoretical guarantees. Moreover, unlike existing best-arm identification strategies, our algorithm uses a stopping rule that does not depend on the number of arms. Experimental results suggest that our algorithm significantly outperforms existing algorithms. The paper further provides a first analysis of the best-arm identification problem in linear bandits with a continuous set of arms.

## 1 Introduction

The stochastic linear bandit [1, 2] is a sequential decision-making problem that generalizes the classical stochastic Multi-Armed Bandit (MAB) problem [3, 4] by assuming that the average reward is a linear function of the arm. Linear bandits have been extensively applied in online services such us online advertisement and recommendation systems [5, 6, 7], and constitute arguably the most relevant *structured* bandit model in practice. Most existing analyses of stochastic linear bandits concern regret minimization [2, 8, 9, 10, 11], i.e., the problem of devising an online algorithm maximizing the expected reward accumulated over a given time horizon. When the set of arms is finite, this problem is *solved* in the sense that we know an instance-specific regret lower bound, and a simple algorithm whose regret matches this fundamental limit [10, 11].

The best-arm identification problem (also referred to as *pure exploration* problem) in linear bandits with finite set of arms has received less attention [12, 13, 14, 15, 16], and does not admit a fully satisfactory solution. In the pure exploration problem with fixed confidence, one has to design a $\delta$-PAC algorithm (able to identify the best arm with probability at least $1 - \delta$) using as few samples as possible. Such an algorithm consists of a sampling rule (an active policy to sequentially select arms), a stopping rule, and a decision rule that outputs the estimated best arm. The number of rounds before the algorithm stops is referred to as its sample complexity. An instance-specific information-theoretical lower bound of the expected sample complexity has been derived in [17]. However, we are lacking simple and practical algorithms achieving this bound. Importantly, existing algorithms exhibit scalability issues as they always include subroutines that explicitly depend on the number of arms (refer to the related work for details). They may also be computationally involved.

In this paper, we present a new best-arm identification algorithm for linear bandits with finite set of arms, whose sample complexity matches the information-theoretical lower bound. The algorithm follows the track-and-stop principle proposed in [18] for pure exploration in bandits without structure. Its sampling rule tracks the optimal proportion of arm draws, predicted by the sample complexity lower bound and estimated using the least-squares estimator of the system parameter. Remarkably,

this tracking procedure can be made as *lazy* as we wish (the estimated optimal proportion of draws can be updated rarely – not every round) without compromising the asymptotic optimality of the algorithm. The stopping rule of our algorithm is classically based on a generalized likelihood ratio test. However the exploration threshold defining its stopping condition is novel, and critically, we manage to make it independent of the number of arms. Overall our algorithm is simple, scalable, and yet asymptotically optimal. In addition, its computational complexity can be tuned by changing the frequency at which the tracking rule is updated, without affecting its theoretical guarantees.

We also study the pure exploration problem in linear bandits with a continuous set of arms. We restrict our attention to the case where the set of arms consists of the $(d-1)$-dimensional unit sphere. We establish a sample complexity lower bound satisfied by any $(\epsilon, \delta)$-PAC algorithm (such algorithms identify an $\epsilon$-optimal arm with probability at least $1 - \delta$). This bound scales as $\frac{d}{\varepsilon} \log(1/\delta)$. We finally propose an algorithm whose sample complexity matches the lower bound order-wise.

**Related work.** Best-arm identification algorithms in linear bandits with a finite set of $K$ arms have been proposed and analyzed in [12, 13, 14, 15, 16]. Soare et al. [12] leverage tools from G-optimal experimental design to devise the $\mathcal{XY}$-adaptive algorithm returning the best arm and with sample complexity $\tau$ satisfying $\tau \lesssim (M^\star \vee T_\mu^\star \log(K^2/\delta))(\log\log(K^2/\delta) + \log(1/\Delta_{\min}^2))$, w.p. $1 - \delta$, where $\mu$ is the parameter defining the reward function, $\Delta_{\min}$ is the minimal gap between the best and a sub-optimal arm, $T_\mu^\star \log(1/\delta)$ is the information theoretical lower bound for the expected sample complexity of $\delta$-PAC algorithms, and where $M^\star$ is an instance-dependent constant. $\mathcal{XY}$-adaptive runs in phases, and eliminates arms at the end of each phase. The use of phases requires rounding procedures, which come with $d^2$ additional rounds in the sample complexity. The algorithm also requires to solve in each round an optimization problem similar to that leading to the sample complexity lower bound (see §3.1). Improved versions of $\mathcal{XY}$-adaptive have been proposed in [15, 16]. ALBA [15] relies on a novel estimator for $\mu$ (removing the need of rounding procedures). RAGE [16] offers an improved sample complexity $\tau \lesssim T_\mu^* \log(1/\Delta_{\min}^2)(\log(K^2/\delta) + d \log(1/\Delta_{\min}^2))$ (slightly simplifying the expression). The aforementioned algorithms are rather complicated, and explicitly use the number $K$ of arms in some of their components: $K$ is present in the arm elimination function in $\mathcal{XY}$-adaptive, in the phase durations in [15, 16]. Importantly, their sample complexity does not match the information-theoretical lower bound when $\delta$ decreases. There is also no guarantees for their expected sample complexity.

[13] proposes an algorithm based on an explore-and-verify framework and with an asymptotically optimal sample complexity. The algorithm is not practical, but is the first to demonstrate that the lower bound derived in [17] is achievable. In [14], the authors present LinGapE, an algorithm, as simple as ours. However, its sampling and stopping rules are both sub-optimal (e.g. the algorithm needs to sample all arms at least once), which in turn leads to weak performance guarantees with a sample complexity satisfying $\tau \lesssim K \log(1/\delta)$.

Recently, Degenne et al. [19] proposed LinGame and LinGame-C, two track-and-stop algorithms that achieve the best possible sample complexity asymptotically. Their stopping rule is somewhat similar to ours, but has the disadvantage of requiring boundedness on the unknown parameter $\mu$. Their sampling rule is different than ours and relies on a game theoretic approach. The latter consists in viewing the information theoretic constant $T_\mu^\star$ as the result of a zero-sum game between the agent and the environment. It is not clear how computationally efficient this algorithm is as it requires solving a saddle-point problem at each round. Finally, we would like to mention that [19] was not available to us at the time of submission.

The algorithm we present is as simple as LinGapE, does not run in phases, does not explicitly use the number of arms in its sampling and stopping rules, and has an asymptotically optimal sample complexity, both almost surely and in expectation.

We are not aware of any work on best-arm identification in linear bandits with a continuous set of arms. We provide here the first results.

## 2 Model and Objective

We consider a bandit problem with a set $\mathcal{A} \subset \mathbb{R}^d$ of arms. In round $t \geq 1$, if the decision maker selects arm $a$, she observes as a feedback a random reward $r_t = \mu^\top a + \eta_t$. $\mu \in \mathbb{R}^d$ is unknown, and $(\eta_t)_{t \geq 1}$ is a sequence of i.i.d. Gaussian random variables, $\eta_t \sim \mathcal{N}(0, \sigma^2)$. The objective is to

learn the arm $a_\mu^\star$ with the highest expected reward $a_\mu^\star = \arg\max_{a \in \mathcal{A}} \mu^\top a$. Throughout the paper, we assume that $\mu$ and $\mathcal{A}$ are such that the best arm $a_\mu^\star$ is unique. We also assume that the set of arms $\mathcal{A}$ spans $\mathbb{R}^d$.

A best-arm identification algorithm consists of a sampling rule, a stopping rule, and a decision rule. The sampling rule decides which arm $a_t$ is selected in round $t$ based on past observations: $a_t$ is $\mathcal{F}_{t-1}$-measurable, where $\mathcal{F}_t$ is the $\sigma$-algebra generated by $(a_1, r_1, \ldots, a_t, r_t)$. The stopping rule decides when to stop sampling, and is defined by $\tau$, a stopping time w.r.t. the filtration $(\mathcal{F}_t)_{t \geq 1}$. The decision rule outputs a guess $\hat{a}_\tau$ of the best arm based on observations collected up to round $\tau$, i.e., $\hat{a}_\tau$ is $\mathcal{F}_\tau$-measurable. The performance of an identification algorithm is assessed through its probabilistic guarantees, and through its sample complexity $\tau$. We consider different probabilistic guarantees, depending on whether the set of arms $\mathcal{A}$ is finite or continuous. Specifically: for $\epsilon, \delta > 0$,

**Definition 1** (Finite set of arms $\mathcal{A}$). An algorithm is $\delta$-PAC if for all $\mu$, $\mathbb{P}_\mu[\hat{a}_\tau \neq a_\mu^\star] \leq \delta$ and $\mathbb{P}_\mu[\tau < \infty] = 1$.

**Definition 2** (Continuous set of arms $\mathcal{A}$). An algorithm is $(\varepsilon, \delta)$-PAC if for all $\mu$, $\mathbb{P}_\mu[\mu^\top(a_\mu^\star - \hat{a}_\tau) > \varepsilon] \leq \delta$ and $\mathbb{P}_\mu[\tau < \infty] = 1$.

When the set of arms $\mathcal{A}$ is finite (resp. continuous), the objective is to devise a $\delta$-PAC (resp. $(\varepsilon, \delta)$-PAC) algorithm with minimal expceted sample complexity $\mathbb{E}_\mu[\tau]$.

**Notation.** Let $K = |\mathcal{A}|$ when $\mathcal{A}$ is finite. $\Lambda = \{x \in [0,1]^{\mathcal{A}} : \sum_{a \in \mathcal{A}} x_a = 1\}$ denotes the simplex in dimension $K$. For $a, b \in [0,1]$, $\mathrm{kl}(a,b)$ is the KL divergence between two Bernoulli distributions of respective means $a$ and $b$. For any $w, w' \in \mathbb{R}^{\mathcal{A}}$, we denote $d_\infty(w, w') = \max_{a \in \mathcal{A}} |w_a - w'_a|$, and for any compact set $C \subset \mathbb{R}^{\mathcal{A}}$, $d_\infty(w, C) = \min_{w' \in C} d_\infty(w, w')$. For $w \in \mathbb{R}^{\mathcal{A}}$, $\mathrm{supp}(w) = \{a \in \mathcal{A} : w_a \neq 0\}$ denotes the support of $w$. $\mathbb{P}_\mu$ (resp. $\mathbb{E}_\mu$) denotes the probability measure (resp. expectation) of observations generated under $\mu$; in absence of ambiguity, we simply use $\mathbb{P}$ (resp. $\mathbb{E}$). For two functions $f$ and $g$, we write $f \lesssim g$ iff there exists a universal constant $C$ such that for all $x$, $f(x) \leq Cg(x)$.

## 3 Finite set of arms

Consider a finite set $\mathcal{A}$ of $K$ arms. We first recall existing lower bounds on the expected complexity of $\delta$-PAC algorithms, and then present our algorithm along with an analysis of its sample complexity.

### 3.1 Sample complexity lower bound

Soare [17] derived the following sample complexity lower bound, using the method developed by Garivier and Kaufmann [20] in the case of bandits without structure.

**Theorem 1.** The sample complexity of any $\delta$-PAC algorithm satisfies: $\forall \mu$, $\mathbb{E}_\mu[\tau] \geq \sigma^2 T_\mu^\star \mathrm{kl}(\delta, 1 - \delta)$, where

$$(T_\mu^\star)^{-1} = \sup_{w \in \Lambda} \min_{a \in \mathcal{A} \setminus a_\mu^\star} \frac{(\mu^\top(a_\mu^\star - a))^2}{2(a_\mu^\star - a)^\top \left(\sum_{a \in \mathcal{A}} w_a a a^\top\right)^{-1} (a_\mu^\star - a)}. \qquad (1)$$

In the above lower bound, $w$ may be interpreted as the proportions of arm draws, also referred to as *allocation*. For $a \in \mathcal{A}$, $w_a$ represents the fraction of rounds where arm $a$ is selected. This interpretation stems from the proof of Theorem 1, where $w_a = \mathbb{E}_\mu[N_a(\tau)]/\mathbb{E}_\mu[\tau]$ and $N_a(t)$ is the number of times $a$ is selected up to and including round $t$ (see [17]). The lower bound is obtained by taking the supremum over $w$, i.e., over the best possible allocation.

A different way to define $(T_\mu^\star)^{-1}$ is $\sup_{w \in \Lambda} \psi(\mu, w)$ (a convex program) [17], where

$$\psi(\mu, w) = \min_{\{\lambda : \exists a \neq a_\mu^\star, \lambda^\top(a^\star - a) < 0\}} \frac{1}{2}(\mu - \lambda)^\top \left(\sum_{a \in \mathcal{A}} w_a a a^\top\right) (\mu - \lambda). \qquad (2)$$

The next lemmas, proved in Appendix B, confirm that the two definitions of $T_\mu^\star$ are equivalent, and provide useful properties of the function $\psi$ and of its maximizers.

**Lemma 1.** We have:

$$\psi(\mu, w) = \begin{cases} \min_{a \in \mathcal{A} \setminus a_\mu^\star} \frac{\langle \mu, a_\mu^\star - a \rangle^2}{2(a_\mu^\star - a)^\top \left( \sum_{i=1}^{K} w_i a_i a_i^\top \right)^{-1} (a_\mu^\star - a)} & \text{if } \sum_{a \in \mathcal{A}} w_a a a^\top \succ 0, \\ 0 & \text{otherwise.} \end{cases} \quad (3)$$

In addition, $\psi$ is continuous in both $\mu$ and $w$, and $w \mapsto \psi(\mu, w)$ attains its maximum in $\Lambda$ at a point $w_\mu^\star$ such that $\sum_{a \in \mathcal{A}} (w_\mu^\star)_a a a^\top$ is invertible.

**Lemma 2.** (Maximum theorem) Let $\mu \in \mathbb{R}^d$ such that $a_\mu^\star$ is unique. Define $\psi^*(\mu) = \max_{w \in \Lambda} \psi(\mu, w)$ and $C^\star(\mu) = \arg \max_{w \in \Lambda} \psi(\mu, w)$. Then $\psi^\star$ is continuous at $\mu$, and $C^\star(\mu)$ is convex, compact and non-empty. Furthermore, we have[1] for any open neighborhood $\mathcal{V}$ of $C^\star(\mu)$, there exists an open neighborhood $\mathcal{U}$ of $\mu$, such that for all $\mu' \in \mathcal{U}$, we have $C^\star(\mu') \subseteq \mathcal{V}$.

### 3.2 Least-squares estimator

Our algorithm and its analysis rely on the least-squares estimator of $\mu$ and on its performance. This estimator $\hat{\mu}_t$ based on the observations in the $t$ first rounds is: $\hat{\mu}_t = (\sum_{s=1}^{t} a_s a_s^\top)^\dagger (\sum_{s=1}^{t} a_s r_s)$. The following result provides a sufficient condition on the sampling rule for the convergence of $\hat{\mu}_t$ to $\mu$. This condition depends on the asymptotic spectral properties of the covariates matrix $\sum_{s=1}^{t} a_s a_s^\top$. We also provide a concentration result for the least-squares estimator. Refer to Appendix C for the proofs of the following lemmas.

**Lemma 3.** Assume that the sampling rule satisfies $\liminf_{t \to \infty} \lambda_{\min} \left( \frac{1}{t^\alpha} \sum_{s=1}^{t} a_s a_s^\top \right) > 0$ almost surely (a.s.), for some $\alpha \in (0, 1)$. Then, $\lim_{t \to \infty} \hat{\mu}_t = \mu$ a.s.. More precisely, for all $\beta \in (0, \alpha/2)$, $\|\hat{\mu}_t - \mu\| = o(t^{-\beta})$ a.s..

**Lemma 4.** Let $\alpha > 0$ and $L = \max_{a \in \mathcal{A}} \|a\|$. Assume that $\lambda_{\min}(\sum_{s=1}^{t} a_s a_s^\top) \geq ct^\alpha$ a.s. for all $t \geq t_0$ for some $t_0 \geq 0$ and some constant $c > 0$. Then

$$\forall t \geq t_0 \qquad \mathbb{P}\left( \|\hat{\mu}_t - \mu\| \geq \varepsilon \right) \leq (c^{-1/2} L)^d t^{\frac{(1-\alpha)d}{2}} \exp\left( -\frac{c\varepsilon^2 t^\alpha}{4\sigma^2} \right). \quad (4)$$

The least-squares estimator is used in our decision rule. After the algorithm stops in round $\tau$, it returns the arm $\hat{a}_\tau \in \arg \max_{a \in \mathcal{A}} \hat{\mu}_\tau^\top a$.

### 3.3 Sampling rule

To design an algorithm with minimal sample complexity, the sampling rule should match optimal proportions of arm draws, i.e., an allocation in the set $C^\star(\mu)$. Since $\mu$ is unknown, our sampling rule will track, in round $t$, allocations in the plug-in estimate $C^\star(\hat{\mu}_t)$. To successfully apply this *certainty equivalence* principle, we need to at least make sure that using our sampling rule, $\hat{\mu}_t$ converges to $\mu$. Using Lemma 3, we can design a family of sampling rules with this guarantee:

**Lemma 5.** (Forced exploration) Let $\mathcal{A}_0 = \{a_0(1), \ldots, a_0(d)\} \subseteq \mathcal{A} : \lambda_{\min}(\sum_{a \in \mathcal{A}_0} a a^\top) > 0$. Let $(b_t)_{t \geq 1}$ be an arbitrary sequence of arms. Furthermore, define for all $t \geq 1$, $f(t) = c_{\mathcal{A}_0} \sqrt{t}$ where $c_{\mathcal{A}_0} = \frac{1}{\sqrt{d}} \lambda_{\min} \left( \sum_{a \in \mathcal{A}_0} a a^\top \right)$. Consider the sampling rule, defined recursively as: $i_0 = 1$, and for $t \geq 0$, $i_{t+1} = (i_t \mod d) + \mathbb{1}_{\{\lambda_{\min}(\sum_{s=1}^{t} a_s a_s^\top) < f(t)\}}$ and

$$a_{t+1} = \begin{cases} a_0(i_t) & \text{if } \lambda_{\min}\left( \sum_{s=1}^{t} a_s a_s^\top \right) < f(t), \\ b_t & \text{otherwise.} \end{cases} \quad (5)$$

Then for all $t \geq \frac{5d}{4} + \frac{1}{4d} + \frac{3}{2}$, we have $\lambda_{\min} \left( \sum_{s=1}^{t} a_s a_s^\top \right) \geq f(t - d - 1)$.

A sampling rule of the family defined in Lemma 5 is forced to explore an arm in $\mathcal{A}_0$ (in a round robin manner) if $\lambda_{\min}(\sum_{s=1}^{t} a_s a_s^\top)$ is too small. According to Lemma 3, this forced exploration is enough to ensure that $\hat{\mu}_t$ converges to $\mu$ a.s.. Next in the following tracking lemma, we show how to design the sequence $(b_t)_{t \geq 1}$ so that the sampling rule gets close to a set $C$ we wish to track.

**Lemma 6.** (Tracking a set $C$) Let $(w(t))_{t\geq 1}$ be a sequence taking values in $\Lambda$, such that there exists a compact, convex and non empty subset $C$ in $\Lambda$, there exists $\varepsilon > 0$ and $t_0(\varepsilon) \geq 1$ such that $\forall t \geq t_0, d_\infty(w(t), C) \leq \varepsilon$.
Define for all $a \in \mathcal{A}$, $N_a(0) = 0$. Consider a sampling rule defined by (5) and

$$b_t = \underset{a\in\mathrm{supp}(\sum_{s=1}^t w(s))}{\arg\min} \left( N_a(t) - \sum_{s=1}^t w_a(s) \right), \tag{6}$$

where $N_a(0) = 0$ and for $t \geq 0$, $N_a(t+1) = N_a(t) + \mathbb{1}_{\{a_t=a\}}$.
Then there exists $t_1(\varepsilon) \geq t_0(\varepsilon)$ such that $\forall t \geq t_1(\varepsilon), d_\infty((N_a(t)/t)_{a\in\mathcal{A}}, C) \leq (p_t + d - 1)\varepsilon$ where $p_t = |\mathrm{supp}(\sum_{s=1}^t w(s))\backslash \mathcal{A}_0| \leq K - d$.

**The lazy tracking rule.** The design of our tracking rule is completed by choosing the sequence $(w(t))_{t\geq 1}$ in (6). The only requirement we actually impose on this sequence is the following condition: there exists a non-decreasing sequence $(\ell(t))_{t\geq 1}$ of integers with $\ell(1) = 1$, $\ell(t) \leq t - 1$ for $t > 1$ and $\lim_{t\to\infty} \ell(t) = \infty$ and such that

$$\lim_{t\to\infty} \min_{s\geq\ell(t)} d_\infty(w(t), C^\star(\hat{\mu}_s)) = 0. \qquad a.s.. \tag{7}$$

This condition is referred to as the *lazy* condition, since it is very easy to ensure in practice. For example, it holds for the following lazy tracking rule. Let $\mathcal{T} = \{t_n : n \geq 1\}$ be a deterministic increasing set of integers such that $t_n \to \infty$ as $n \to \infty$, we can define $(w(t))_{t\geq 1}$ such that it tracks $C^\star(\hat{\mu}_t)$ only when $t \in \mathcal{T}$. Specifically, if $t \in \mathcal{T}$, $w(t+1) \in C^\star(\hat{\mu}_t)$, and $w(t+1) = w(t)$ otherwise. For this sequence, (7) holds with $\ell(t) = t - 1$. The lazy condition is sufficient to guarantee the almost sure asymptotical optimality of the algorithm. To achieve optimality in terms of expected sample complexity, we will need a slightly stronger condition, also easily satisfied under some of the above lazy tracking rules, see details in §3.5.

The following proposition states that the lazy sampling rule is able to track the set $C^*(\mu)$. It follows from the fact that $\hat{\mu}_t$ converges to $\mu$ (thanks to Lemmas 3 and 5) and from combining the maximum theorem (Lemma 2) and Lemma 6. All proofs related to the sampling rule are presented in Appendix E.

**Proposition 1.** Under any sampling rule (5)-(6) satisfying the lazy condition (7), the proportions of arm draws approach $C^\star(\mu)$: $\lim_{t\to\infty} d_\infty((N_a(t)/t)_{a\in\mathcal{A}}, C^\star(\mu)) = 0$, a.s..

### 3.4 Stopping rule

We use the classical Chernoff's stopping time. Define the generalized log-likelihood ratio for all pair of arms $a, b \in \mathcal{A}$, $t \geq 1$, and $\varepsilon \geq 0$ as

$$Z_{a,b,\varepsilon}(t) = \log\left( \frac{\max_{\{\mu:\mu^\top(a-b)\geq-\varepsilon\}} f_\mu(r_t, a_t, \ldots, r_1, a_1)}{\max_{\{\mu:\mu^\top(a-b)\leq-\varepsilon\}} f_\mu(r_t, a_t, \ldots, r_1, a_1)} \right),$$

where $f_\mu(r_t, a_t, \ldots, r_1, a_1) \propto \exp(-\frac{1}{2}\sum_{s=1}^t (r_s - \mu^\top a_s)^2)$ under our Gaussian noise assumption. We may actually derive an explicit expression of $Z_{a,b,\varepsilon}(t)$ (see Appendix D for a proof):

**Lemma 7.** Let $t \geq 1$ and assume that $\sum_{s=1}^t a_s a_s^\top \succ 0$. For all $a, b \in \mathcal{A}$, we have:

$$Z_{a,b,\varepsilon}(t) = \mathrm{sgn}(\hat{\mu}_t^\top(a-b) + \varepsilon) \frac{(\hat{\mu}_t^\top(a-b) + \varepsilon)^2}{2(a-b)^\top \left( \sum_{s=1}^t a_s a_s^\top \right)^{-1} (a-b)}.$$

Here we use $Z_{a,b}(t) = Z_{a,b,0}(t)$ ($Z_{a,b,\varepsilon}$ will be used in the case of continuous set of arms). Note that $Z_{a,b}(t) \geq 0$ iff $a \in \arg\max_{a\in\mathcal{A}} \hat{\mu}_t^\top a$. Denoting $Z(t) = \max_{a\in\mathcal{A}} \min_{b\in\mathcal{A}\backslash a} Z_{a,b}(t)$, the stopping rule is defined as follows:

$$\tau = \inf\left\{ t \in \mathbb{N}^* : Z(t) > \beta(\delta, t) \text{ and } \sum_{s=1}^t a_s a_s^\top \succeq cI_d \right\} \tag{8}$$

where $\beta(\delta, t)$ is referred to as the *exploration threshold* and $c$ is some positive constant (refer to Remark 1 for a convenient choice for $c$). The exploration threshold $\beta(\delta, t)$ should be chosen so that

the algorithm is $\delta$-PAC. We also wish to design a threshold that does not depend in the number $K$ of arms. These requirements leads to the exploration threshold defined in the proposition below (its proof is presented in Appendix D and relies on a concentration result for self-normalized processes [9]).

**Proposition 2.** Let $u > 0$, and define:

$$\beta(\delta, t) = (1 + u)\sigma^2 \log \left( \frac{\det \left( (uc)^{-1} \sum_{s=1}^t a_s a_s^\top + I_d \right)^{\frac{1}{2}}}{\delta} \right). \qquad (9)$$

Under any sampling rule, and a stopping rule (8) with exploration rate (9), we have: $\mathbb{P}\left( \tau < \infty, \mu^\top (a_\mu^\star - \hat{a}_\tau) > 0 \right) \leq \delta$.

The above proposition is valid for any sampling rule, but just ensures that 'if' the algorithm stops, it does not make any mistake w.p. $1 - \delta$. To get a $\delta$-PAC algorithm, we need to specify the sampling rule.

**Remark 1.** (Choosing $c$ and $u$) A convenient choice for the constant $c$ involved in (8) and (9) is $c = \max_{a \in \mathcal{A}} \|a\|^2$. With this choice, we have: $\det(c^{-1} \sum_{s=1}^t a_s a_s^\top + I_d) \leq (t+1)^d$. The constant $u$ should be chosen so that the threshold in (9) is lowered for instance one my choose $u = 1$. From these choices the threshold can be as simple as $\beta(\delta, t) = 2\sigma^2 \log(t^{\frac{d}{2}}/\delta)$. In addition, if we use a sampling rule with forced exploration as in (5), then in view of Lemma 5, the second stopping condition $\sum_{s=1}^t a_s a_s^\top \succeq cI_d$ is satisfied as soon as $t$ exceeds $d + 1 + \frac{c^2 d}{\lambda_{\min}(\sum_{a \in \mathcal{A}_0} aa^\top)}$.

## 3.5 Sample complexity analysis

---
**Algorithm 1:** Lazy Track-and-Stop (LT&S)

---
**Input:** Arms $\mathcal{A}$, confidence level $\delta$, set $\mathcal{T}$ of lazy updates
**Initialization:** $t = 0$, $i = 0$, $A_0 = 0$, $Z(0) = 0$, $N(0) = (N_a(0))_{a \in \mathcal{A}} = 0$;
**while** $(\lambda_{\min}(A_t) < c)$ *or* $(Z(t) < \beta(\delta, t))$ **do**
    **if** $\lambda_{\min}(A_t) < f(t)$ **then**
        $a \leftarrow a_0(i+1)$, $i \leftarrow (i+1 \mod d)$
    **else**
        $a \leftarrow \arg\min_{b \in \text{supp}(\sum_{s=1}^t w(s))} \left( N_b(t) - \sum_{s=1}^t w_b(s) \right)$,
    **end**
    $t \leftarrow t + 1$, sample arm $a$ and update $N(t)$, $\hat{\mu}_t$, $Z(t)$, $A_t \leftarrow A_{t-1} + aa^\top$, $w(t) \leftarrow w(t-1)$
    **if** $t \in \mathcal{T}$ **then**
        $w(t) = \arg\max_{w \in \Lambda} \psi(\hat{\mu}_t, w)$
    **end**
**end**
**return** $\hat{a}_\tau = \arg\max_{a \in \mathcal{A}} \hat{\mu}_\tau^\top a$

---

In this section, we establish that combining a sampling rule (5)-(6) satisfying the lazy condition (7) and the stopping rule (8)-(9), we obtain an asymptotically optimal algorithm. Refer to Appendix F for proofs. An example of such algorithm is the Lazy Track-and-Stop (LT&S) algorithm, whose pseudo-code is presented in Algorithm 1. LT&S just updates the tracking rule in rounds in a set $\mathcal{T}$.

**Theorem 2.** (Almost sure sample complexity upper bound) An algorithm defined by (5)-(6)-(8)-(9) with a lazy sampling rule (satisfying (7)) is $\delta$-PAC. Its sample complexity verifies:

$$\mathbb{P}(\limsup_{\delta \to 0} \frac{\tau}{\log(\frac{1}{\delta})} \lesssim \sigma^2 T_\mu^*) = 1.$$

To obtain an algorithm with optimal expected sample complexity, we need to consider lazy tracking rules that satisfy the following condition: there exist $\alpha > 0$ and a non-decreasing sequence $(\ell(t))_{t \geq 1}$ of integers with $\ell(1) = 1$, $\ell(t) \leq t$ and $\liminf_{t \to \infty} \ell(t)/t^\gamma > 0$ for some $\gamma > 0$ and such that

$$\forall \varepsilon > 0, \ \exists h(\varepsilon) \ : \ \forall t \geq 1, \ \mathbb{P}\left( \min_{s \geq \ell(t)} d_\infty(w(t), C^\star(\hat{\mu}_s)) > \varepsilon \right) \leq \frac{h(\varepsilon)}{t^{2+\alpha}}. \qquad (10)$$

The condition (10) is again easy to ensure. Assume that we update $w(t)$ only if $t \in \mathcal{T} = \{t_n : n \geq 1\}$, where $t_n$ is increasing sequence of integers such that $t_n \to \infty$ as $n \to \infty$. Then (10) holds for the sequence $(\ell(t))_{t \geq 1}$ such that $\ell(t_{i+1}) = t_i$ for all $i$, provided that $\liminf_{n \to \infty} t_{n+1}/t_n^{\gamma} > 0$ for some $\gamma > 0$. Examples include: (i) periodic updates of $w(t)$: $\mathcal{T} = \{1 + kP, k \in \mathbb{N}\}$ and $\ell(t) = \max\{1, t - P\}$; (ii) exponential updates $\mathcal{T} = \{2^k, k \in \mathbb{N}\}$ and $\ell(t) = \max\{1, \lfloor t/2 \rfloor\}$. The condition (10) may seem too loose, but we have to keep in mind that in practice, the performance of the algorithm will depend on the update frequency of $w(t)$. However for asymptotic optimality, (10) is enough (the key point is to have some concentration of $\hat{\mu}_t$ around $\mu$, which is guaranteed via the forced exploration part of the sampling rule).

**Theorem 3.** (Expected sample complexity upper bound) An algorithm defined by (5)-(6)-(8)-(9) with a sampling rule satisfying (7) and (10) is $\delta$-PAC. Its sample complexity verifies:

$$\limsup_{\delta \to 0} \frac{\mathbb{E}[\tau]}{\log\left(\frac{1}{\delta}\right)} \lesssim \sigma^2 T_\mu^\star.$$

# 4 Continuous set of arms

We now investigate the case where $\mathcal{A} = S^{d-1}$ is the $(d-1)$-dimensional unit sphere. Without loss of generality, we restrict our attention to problems where $\mu \in \mathcal{M}(\varepsilon_0) = \{\eta : \eta^\top a_\eta^\star > \varepsilon_0\}$ for some $\varepsilon_0 > 0$. The results of this section are proved in Appendix G.

## 4.1 Sample complexity lower bound

**Theorem 4.** Let $\varepsilon \in (0, \varepsilon_0/5)$, and $\delta \in (0, 1)$. The sample complexity of any $(\delta, \varepsilon)$-PAC algorithm satisfies: for all $\mu \in \mathcal{M}(\varepsilon_0)$, $\mathbb{E}_\mu[\tau] \geq \frac{\sigma^2(d-1)}{20\|\mu\|\varepsilon} \mathrm{kl}(\delta, 1 - \delta)$.

The above theorem is obtained by first applying the classical change-of-measure argument (see e.g. Lemma 19 [20]). Such an argument implies that under any $(\varepsilon, \delta)$-PAC algorithm, for all *confusing* $\lambda$ such that $\{a \in S^{d-1} : \mu^\top(a_\mu^\star - a) \leq \varepsilon\}$ and $\{a \in S^{d-1} : \lambda^\top(a_\lambda^* - a) \leq \varepsilon\}$ are disjoint,

$$(\mu - \lambda)^\top \mathbb{E}\left[\sum_{s=1}^{\tau} a_s a_s^\top\right] (\mu - \lambda) \geq 2\mathrm{kl}(\delta, 1 - \delta).$$

We then study the solution of the following max-min problem: $\max_{(a_t)_{t \geq 1}} \min_{\lambda \in B_\varepsilon(\mu)} (\mu - \lambda)^\top \mathbb{E}\left[\sum_{s=1}^{\tau} a_s a_s^\top\right] (\mu - \lambda)$, where $B_\varepsilon(\mu)$ denotes the set of confusing parameters. The continuous action space makes this analysis challenging. We show that the value of the max-min problem is smaller than $\mathbb{E}_\mu[\tau] \frac{10\|\mu\|\varepsilon}{\sigma^2(d-1)}$, which leads to the claimed lower bound.

## 4.2 Algorithm

We present a simple algorithm whose sample complexity approach our lower bound. We describe its three components below. The decision rule is the same as before, based on the least-squares estimator of $\mu$: $\hat{a}_t \in \arg\max_{a \in \mathcal{A}} \hat{\mu}_t^\top a$.

**Sampling rule.** Let $\mathcal{U} = \{u_1, u_2, \ldots, u_d\}$ be subset of $S^{d-1}$, that forms an orthonormal basis of $\mathbb{R}^d$. The sampling rule just consists in selecting an arms from $\mathcal{U}$ in a round robin manner: for all $t \geq 1$, $a_t = u_{(t \mod d)}$.

**Stopping rule.** As for the case of finite set of arms, the stopping rule relies on a generalized loglikelihood ratio test. Define $Z(t) = \inf_{\{b \in \mathcal{A}: |\hat{\mu}_t^\top(\hat{a}_t - b)| \geq \varepsilon_t\}} Z_{\hat{a}_t, b, \varepsilon_t}(t)$, where an expression of $Z_{\hat{a}_t, b, \varepsilon_t}(t)$ is given in Lemma 7. We consider the following stopping time:

$$\tau = \inf\left\{t \in \mathbb{N}^* : Z(t) \geq \beta(\delta, t) \text{ and } \lambda_{\min}\left(\sum_{s=1}^{t} a_s a_s^\top\right) \geq \max\left\{c, \frac{\rho(\delta, t)}{\|\hat{\mu}_t\|^2}\right\}\right\}. \quad (11)$$

Hence compared to the case of finite set of arms, we add a stopping condition defined by the threshold $\rho(\delta, t)$ and related to the spectral properties of the covariates matrix.

**Proposition 3.** Let $(\delta_t)_{t\geq 1}, (\varepsilon_t)_{t\geq 1}$ be two sequences with values in $(0,1)$ and $(0,\varepsilon)$, respectively, and such that $\sum_{t=1}^\infty \delta_t < \delta$, and $\lim_{t\to\infty} \varepsilon_t = \varepsilon$. Let $\zeta_t = \log(2\det\left(c^{-1}\sum_{s=1}^t a_s a_s^\top + I_d\right)^{\frac{1}{2}}) - \log(\delta_t)$, and define:

$$\beta(\delta, t) = 2\sigma^2 \zeta_t \quad \text{and} \quad \rho(\delta, t) = \frac{4\sigma^2 \varepsilon_t^2 \zeta_t}{(\varepsilon - \varepsilon_t)^2} \tag{12}$$

Then under the stopping rule (11)-(12), we have: $\mathbb{P}_\mu\left(\tau < \infty, \mu^\top(a_\mu^\star - \hat{a}_\tau) > \varepsilon\right) \leq \delta$.

### 4.3 Sample complexity analysis

Under specific choices for the sequence $(\varepsilon_t)_{t\geq 1}$, we can analyze the sample complexity of our algorithm, and show its optimality order-wise.

**Theorem 5.** Choose in the stopping rule $\varepsilon_t = \varepsilon\left(1 + \varepsilon(4\sigma^2 \log(\frac{4}{\delta_t}\lceil\frac{t}{d}\rceil))^{-1/2}\right)^{-1}$ (observe that $\varepsilon_t < \varepsilon$ and $\lim_{t\to\infty}\varepsilon_t = \varepsilon$). Then under the aforementioned sampling rule, and the stopping rule (11)-(12), we have: $\mathbb{P}\left(\limsup_{\delta\to 0}\frac{\tau}{\log(1/\delta)} \lesssim \frac{\sigma^2 d}{\|\mu\|\varepsilon}\right) = 1$ and $\limsup_{\delta\to 0}\frac{\mathbb{E}[\tau]}{\log(1/\delta)} \lesssim \frac{\sigma^2 d}{\|\mu\|\varepsilon}$.

## 5 Experiments

We present here a few experimental results comparing the performance of our algorithm to that of RAGE, the state-of-the-art algorithm [16], in the case of finite set of arms. We compare LT&S and RAGE only because they outperform other existing algorithms. Further experimental results can be found in Appendix A.

**Experimental set-up.** We use the following toy experiment which corresponds to the many arms example in [16]. $d = 2$ and $\mathcal{A} = \{(1,0), e^{j3\pi/4}, e^{j(\pi/4+\phi_i)}, i \in [n-2]\} \subset \mathbb{C}$ where $(\phi_i)$ are i.i.d. $\sim \mathcal{N}(0, 0.09)$. $\mu = (1,0)$. Experiments are made with the risk $\delta = 0.05$.

**Implementation of LT&S.** To update the allocation $w(t)$, we use Frank-Wolfe algorithm [21] (without any rounding procedure). Our implementation of Frank-Wolfe is similar to that of Fiez et al. [16] (refer to Appendix A.3. for further comments). At each update, the previous allocation is fed as an initial value for the new optimization problem. We implement the exponential lazy update scheme $\mathcal{T} = \{2^k, k \in \mathbb{N}\}$. The parameters of our stopping rule are $c = c_{\mathcal{A}_0}\sqrt{d}$ (so that after $d$ steps the second condition of the stopping rule is satisfied) and $u = 1$; we use the threshold $\beta(6\delta/\pi^2 t^2, t)$. The initial exploration matrix $\mathcal{A}_0$ is chosen at random. We implemented two versions of LT&S. The first one does not track the average but only the current allocation $w(t)$: $a \leftarrow \arg\min_{b\in\text{supp}(w(t))}(N_b(t) - tw_b(t))$. The second version tracks the average allocations as described in Algorithm 1.

We further compare our results to that of the Oracle algorithm proposed by [12]. The algorithm samples from a true optimal allocation $w^\star \in C^\star(\mu)$, and applies a stopping rule that depends on $K$.

**Results.** From the table below, LT&S outperforms RAGE most of the times, and the performance improvement gets higher when the number of arms grows. LT&S without averaging shows better performance, than with averaging. In Appendix A, we present results for another version of LT&S, with even better performance.

| Algorithm | LT&S (No averaging) Sample Complexity | | LT&S Sample Complexity | | RAGE Sample Complexity | | Oracle Sample Complexity | |
|---|---|---|---|---|---|---|---|---|
| Number of arms | Mean | (Std) | Mean | (Std) | Mean | (Std) | Mean | (Std) |
| $(K = 1000)$ | 1206.55 | (42.2) | 1409 | (57) | **1148.45** | (49.82) | 476.45 | (40.74) |
| $(K = 2500)$ | **1253.60** | (47.70) | 1404 | (57) | 1440.75 | (149.24) | 492.15 | (43.88) |
| $(K = 5000)$ | **1247.05** | (81.07) | 1401 | (86) | 1540.3 | (158.90) | 515.60 | (47.64) |
| $(K = 7500)$ | **1296.55** | (76.78) | 1434 | (78) | 1598.0 | (164.60) | 547.65 | (45.77) |

Table 1: Results for the many arms experiment [16]

# 6 Conclusion

In this paper, we present LT&S, an algorithm to solve the best-arm identification problem in stochastic linear bandits. The sampling rule of the algorithm just tracks the optimal allocation predicted by the sample complexity lower bound. Its stopping rule is defined through generalized log-likelihood ratio and an exploration threshold that does not depend on the number of arms, but on the ambient dimension only. LT&S is asymptotically optimal: we have guarantees on its sample complexity, both almost surely and in expectation. The first experimental results are very promising, as LT&S seems to exhibit a much better sample complexity than existing algorithms. We also provide the first results on the pure exploration problem in the linear bandits with a continuous set of arms.

The analysis presented in this paper suggests several extensions. We can easily generalize the results to non-Gaussian reward distributions (e.g. bounded, from a one-parameter exponential family). It would be interesting to extend our results in the continuous setting to generic convex sets of arms (we believe that the instance-specific sample complexity lower bound would just depend on the local smoothness of the set of arms around the best arm). A more challenging but exciting question is to derive tight non-asymptotic sample complexity upper bound for LT&S, so as to characterize the trade-off between the laziness of the algorithm and its sample complexity.

## Broader impact

This work is mostly theoretical. Our results may provide guidelines and insights towards an improved design of algorithms for linear bandits. Linear bandit algorithms are versatile, and in particular used in clinical trials and recommendation systems. Hence our results can benefit users and developers of such systems. In clinical trials, minimizing the sample complexity is crucial, and the use of our algorithm there can be really beneficial. Our algorithm has the additional advantage of being computationally efficient. Overall, we do not foresee any direct negative impact of our work. However, it worth noting that there is some concern about the potential use of recommender systems for opinion influence.

## Acknowledgement

This research was supported by the Wallenberg AI, Autonomous Systems and Software Program (WASP) funded by the Knut and Alice Wallenberg Foundation.

## Footnotes

[1] This statement is that of upper hemicontinuity of a correspondence.

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
