[Supplementary Material]

# A Numerical experiments

This section provides additional numerical results, and comparisons of LT&S and RAGE. We actually present the results of a slightly different version of LT&S than that considered in the main document (see details below). This new version exhibits much better performance.

## A.1 Experimental set-up

**The benchmark example.** We consider the example proposed by Soare et al. [12] and that has become a standard benchmark to compare best arm identification algorithms in stochastic linear bandits [19, 16, 15, 14]. In this example, the action set is $\mathcal{A} = \{e_1, e_2, \ldots, e_d, a'\}$ where $\{e_1, \ldots, e_d\}$ correspond to the standard basis in $\mathbb{R}^d$, and $a' = \cos(\omega)e_1 + \sin(\omega)e_2$ where the angle $\omega = 0.1$. The unkown parameter $\mu = 2e_1$. For this example, we consider two experiments: (i) we fix $\delta = 0.01$ and vary the dimension $d \in \{2, 3, 4, 5, 6, 7, 8, 9, 10\}$. (ii) we fix $d = 6$ and vary $\delta \in \{0.5, 0, 1, 0.05, 0, 01, 0.005, 0.001\}$.

**The many arms example.** We use the same parameters as those reported in the main document. Namely, the following toy experiment that corresponds to the many arms example in [16]. $d = 2$ and $\mathcal{A} = \{(1,0), e^{j3\pi/4}, e^{j(\pi/4+\phi_i)}, i \in [n-2]\} \subset \mathbb{C}$ where $(\phi_i)$ are i.i.d. $\sim \mathcal{N}(0, 0.09)$. $\mu = (1, 0)$. Experiments are made with the risk $\delta = 0.05$.

**Implementation of LT&S.** Our implementation for the following results is almost the same as the one described in Section 5. The only difference lies in the stopping rule: we use improved constants when defining the threshold (9). The new constant is $u = 0.1$ (before it was set to 1), and the threshold is $\beta(\delta, t)$ (before we were using $\beta(\delta 6/(\pi t)^2, t)$). For the benchmark example, we further consider the following threshold $\tilde{\beta}(\delta, t) = (1 + u)\sigma^2(\log(1/\delta) + 0.5\log(t) + d\log((u^{-1} + 1)^{1/2}))$ for the stopping rule. This threshold is not theoretically proven, but we conjecture that there exists a threshold of the form $c_1 \log(t/\delta) + c_2 d$ with some absolute constants $c_1, c_2 > 0$ such that LT&S is $\delta$-PAC and asymptotically optimal.

All experiments were executed on a stationary desktop computer, featuring an Intel Xeon Silver 4110 CPU, 48GB of RAM. Ubuntu 18.04 was installed on the computer. We set up our experiments using Python 3.7.7.

## A.2 Results

**Sample complexity.** The results on the sample complexity for the many arms experiment are reported in Table 2 and those of the benchmark example are reported in Table 5 and 6. For the many arms experiment, LT&S with and without averaging significantly outperforms RAGE [16] and even the Oracle [12]. At first, it seems surprising that the Oracle is beaten by LT&S, but this can be explained as follows. Even if the Oracle is aware, from the beginning, of the optimal sampling rule, its stopping rule is not efficient and depends on the number of arms $K$. The threshold $\beta(\delta, t)$ in the stopping rule of LT&S is independent of $K$, and indeed, the performance of the algorithm is less sensitive to the number of arms than that of RAGE or the Oracle. The results also suggest that the LT&S algorithms with or without averaging perform similarly. For the benchmark example, LT&S is overall competitive with RAGE and $\mathcal{X}\mathcal{Y}$-adaptive and performs better than ALBA. The results of experiment (i) indicate that the sample complexity of LT&S is affected by the increase of the dimension. On the other hand, LT&S with the modified threshold $\tilde{\beta}(\delta, t)$ performs much better than RAGE. The results of experiment (ii) clearly suggest that the sample complexity of LT&S becomes better as $\delta \to 0$ and is comparable to RAGE and $\mathcal{X}\mathcal{Y}$-Adaptive in the moderate regimes. Again, LT&S with the modified threshold is outperforming all the other algorithms. As a final note all algorithms including LT&S with the modified threshold ended with success in all experiments over all simulations.

**Run-time.** The run-time of LT&S and RAGE are reported in Table 3 for the many arms experiment. Overall, both algorithms are efficient. We note that RAGE is slightly faster. However we expect that for extremely large numbers of arms, LT&S would run faster than RAGE (the sample complexity of LT&S is more resilient to an increase in the number of arms). In LT&S, we have used the exponential lazy update scheme with $\mathcal{T} = \{2^k : k \in \mathbb{N}^*\}$. We believe that by fine-tuning this laziness, we would be able achieve a better trade-off between computational efficiency and sample complexity.

**Support of Lazy T&S.** Finally, we look at the support of the allocation chosen under LT&S. The expected size of the support of LT&S on a single run is reported in Table 4. Even if the number of arms $K$ is large (in comparison with the ambient dimension), LT&S only tracks allocations that are sparse, i.e. using very few arms. We further note that the averaging scheme in the tracking rule does not really affect the support. This is a nice feature as it could allow for the design of a more memory-efficient algorithm.

| Algorithm | LT&S Sample Complexity | | LT&S (No averaging) Sample Complexity | | RAGE Sample Complexity | | Oracle Sample Complexity | |
|---|---|---|---|---|---|---|---|---|
| Number of arms | Mean | (Std) | Mean | (Std) | Mean | (Std) | Mean | (Std) |
| $(K = 1000)$ | **424.5** | (29.1) | **424.5** | (29.1) | 1148.45 | (49.82) | 476.45 | (40.7) |
| $(K = 2500)$ | 458.15 | (28.1) | **455.95** | (28.3) | 1440.75 | (149.24) | 492.15 | (43.9) |
| $(K = 5000)$ | 434.65 | (32.51) | **433.6** | (32.6) | 1540.3 | (158.9) | 515.6 | (47.6) |
| $(K = 7500)$ | 448.0 | (36.9) | **447.45** | (36.8) | 1598.0 | (164.6) | 547.65 | (45.8) |
| $(K = 10000)$ | **452.85** | (31.6) | 452.95 | (31.6) | 1479.4 | (52.0) | 564.85 | (46.9) |

Table 2: Sample complexity. Results for the many arms experiment [16]

| Algorithm | LT&S Run time (s) | | LT&S (No averaging) Run time (s) | | RAGE Rune time (s) | |
|---|---|---|---|---|---|---|
| Number of arms | Mean | (Std) | Mean | (Std) | Mean | (Std) |
| $(K = 1000)$ | 13.62 | (0.5) | 13.99 | (0.5) | 34.0 | (0.5) |
| $(K = 2500)$ | 90.25 | (2.9) | 89.41 | (3.1) | 156.42 | (1.1) |
| $(K = 5000)$ | 940.97 | (40.4) | 948.86 | (40.3) | 429.67 | (7.47) |
| $(K = 7500)$ | 1340.83 | (61.5) | 1349.90 | (61.4) | 707.09 | (9.47) |
| $(K = 10000)$ | 1893.73 | (79.9) | 1915.03 | (80.3) | 1575.30 | (12.43) |

Table 3: Runtime. Results for the many arms experiment [16]

| Algorithm | LT&S Support size | | LT&S (No averaging) Support size | |
|---|---|---|---|---|
| Number of arms | Mean | (Std) | Mean | (Std) |
| $(K = 1000)$ | 5.37 | (0.25) | 2.04 | (0) |
| $(K = 2500)$ | 5.72 | (0.20) | 2.03 | (0) |
| $(K = 5000)$ | 5.41 | (0.20) | 2.04 | (0) |
| $(K = 7500)$ | 5.34 | (0.20) | 2.03 | (0) |
| $(K = 10000)$ | 5.26 | (0.21) | 2.04 | (0) |

Table 4: Support size. Results for the many arms experiment [16]. For the standard deviation, we put $(0)$ when the value is smaller than $10^{-2}$.

| Algorithm | LT&S Sample Complexity | | LT&S (modified threshold) Sample Complexity | | RAGE Sample Complexity | | Oracle Sample Complexity | |
|---|---|---|---|---|---|---|---|---|
| dimension | Mean | (Std) | Mean | (Std) | Mean | (Std) | Mean | (Std) |
| $(d = 2)$ | 3538.69 | (146.9) | **2510.50** | (106.4) | 8185.91 | (591.7) | 3530.95 | (139.8) |
| $(d = 3)$ | 5040.34 | (187.5) | **3107.43** | (134.8) | 7743.22 | (482.3) | 3785.07 | (147.3) |
| $(d = 4)$ | 6236.26 | (209.9) | **3346.29** | (125.3) | 8033.51 | (464.0) | 3968.72 | (163.9) |
| $(d = 5)$ | 7511.91 | (254.1) | **3641.21** | (114.4) | 8796.97 | (511.9) | 3968.27 | (148.9) |
| $(d = 6)$ | 9194.15 | (316.8) | **4087.21** | (136.2) | 8734.26 | (441.0) | 4079.17 | (162.8) |
| $(d = 7)$ | 10321.54 | (325.1) | **4405.74** | (143.5) | 9675.00 | (537.8) | 4107.09 | (160.6) |
| $(d = 8)$ | 11215.20 | (418.1) | **4983.38** | (179.1) | 10025.60 | (550.9) | 4215.97 | (167.5) |
| $(d = 9)$ | 12700.16 | (436.4) | **5078.80** | (169.5) | 10475.19 | (555.5) | 4274.95 | (167.9) |
| $(d = 10)$ | 14049.75 | (463.1) | **5602.55** | (180.9) | 9780.54 | (360.0) | 4321.84 | (167.8) |

Table 5: Sample complexity. Results for experiment (i) of the benchmark example

| Algorithm | LT&S Sample Complexity | | LT&S (modified threshold) Sample Complexity | | RAGE Sample Complexity | | XY Adaptive Sample Complexity | | ALBA Sample Complexity | | Oracle Sample Complexity | |
|---|---|---|---|---|---|---|---|---|---|---|---|---|
| dimension | Mean | (Std) | Mean | (Std) | Mean | (Std) | Mean | (Std) | Mean | (Std) | Mean | (Std) |
| $(\delta = 0.5)$ | 7856.16 | (301.7) | **3080.56** | (119.1) | 5840.21 | (373.6) | 6192.34 | (373.8) | 20330.67 | (837.4) | 3016.91 | (133.1) |
| $(\delta = 0.1)$ | 8408.31 | (308.4) | **3538.96** | (125.4) | 7281.7 | (408.5) | 7785.23 | (380.4) | 26535.6 | (986.4) | 3404.78 | (140.2) |
| $(\delta = 0.05)$ | 8641.79 | (322.8) | **3699.84** | (130.1) | 7751.79 | (434.2) | 8167.51 | (368.3) | 28201.37 | (1047.8) | 3610.33 | (146.1) |
| $(\delta = 0.01)$ | 9194.15 | (316.8) | **4087.21** | (136.1) | 8734.26 | (441.0) | 9255.96 | (317.4) | 32661.52 | (984.0) | 4079.17 | (162.7) |
| $(\delta = 0.005)$ | 9404.55 | (315.0) | **4297.23** | (131.8) | 9810.19 | (543.8) | 9278.82 | (315.8) | 35335.0 | (1148.5) | 4219.38 | (165.3) |
| $(\delta = 0.001)$ | 9583.05 | (313.6) | **4744.36** | (133.3) | 9836.42 | (378.0) | 9897.96 | (268.4) | 39303.0 | (1288.8) | 4644.7 | (172.9) |

Table 6: Sample complexity. Results for experiment (ii) of the benchmark example.

## A.3 Additional remarks and implementation details

**Frank-Wolfe algorithm.** Frank-Wolfe algorithm [21] was used to solve the optimization problem $\max_{w \in \Lambda} \psi(\hat{\mu}_t, w)$ whenever $t \in \mathcal{T}$. The algorithm settings were chosen in a similar way as for RAGE [16]. The step-size was set to $2/(2 + k)$ for each iteration $k$. The algorithm stops when the relative change in $w$ with respect to the 2-norm is lower than $0.01$ or if it reaches a maximum of $1000$ iterations. The components of $w$ were thresholded to 0 if they were smaller than $10^{-5}$ and the $w$ was rescaled properly so that $\sum_{a \in \mathcal{A}} w_a = 1$. It is worth mentioning that there are no guarantees on the convergence of this algorithm in our specific setting. This was highlighted by [19]: there, the authors provided a counter-example for which the algorithm does not converge; they also proposed a new heuristic that seems to converge. However, in our experiments, the algorithm always converged.

**Dependency on $K$.** LT&S does not use explicitly the number $K$ of arms in its stopping rule. More precisely the exploration threshold $\beta(\delta, t)$ is independent of $K$. However, in the algorithm, we still need to allocate data structures with a memory size scaling linearly in $K$ (storing $w_t$ and computing $\max_{w \in \Lambda} \psi(\hat{\mu}_t, w)$). We suspect that using a more efficient implementation, we might be able to circumvent this limitation. For instance, our experiments suggest that the allocations $w_t$ are sparse. Furthermore, our analysis is asymptotic in $\delta$, and does not reveal the dependency of low order terms on $K$ in the sample complexity. We conjecture that such dependency is mild and rather negligible if the minimum gap is relatively large. We leave this question for future work.

# B  Properties of $\psi$

## B.1  Proof of Lemma 1

Let $(\mu, w) \in \mathbb{R}^d \times \Lambda$ such that $a_\mu^\star$ is unique. For the first part of the claim we refer to the proof of [17, Theorem 3.1.]. Now let us prove the continuity of $\psi$ at $(\mu, w)$. Consider the set of bad parameters with respect to $\mu$, $B(\mu) \subseteq \mathbb{R}^d$

$$B(\mu) = \left\{ \lambda : \lambda \in \mathbb{R}^d \text{ and } \exists a \in \mathcal{A} \backslash \{a_\mu^\star\} \ \lambda^\top (a - a_\mu^\star) > 0 \right\},$$

and denote

$$f(\mu, \lambda, w) = \frac{1}{2}(\mu - \lambda)^\top \left( \sum_{a \in \mathcal{A}} w_a a a^\top \right) (\mu - \lambda).$$

Let $(\mu_t, w_t)_{t \geq 1}$ be a sequence taking values in $\mathbb{R}^d \times \Lambda$ and converging to $(\mu, w)$. Let $\varepsilon < 1 \wedge \min_{a \in \mathcal{A} \backslash \{a_\mu^\star\}} \frac{\langle \mu, a_\mu^* - a \rangle}{\|a_\mu^\star - a\|}$, and let $t_1 \geq 1$ such that for all $t \geq t_1$ we have $\|(\mu_t, w_t) - (\mu, w)\| < \varepsilon$. Now, by our choice of $\varepsilon$, and uniqueness of $a_\mu^\star$ it holds that $B(\mu_t) = B(\mu)$. Furthermore, note that $f(\mu, \lambda, w)$ is a polynomial in $\mu, \lambda, w$, thus it is in inparticular continuous in $\mu, w$, and there exists $t_2 \geq 1$ such that for all $t \geq t_2$ and for all $\lambda \in \mathbb{R}^d$, it holds that $|f(\mu_t, \lambda, w_t) - f(\mu, \lambda, \mu_t)| \leq \varepsilon f(\mu, \lambda, \mu_t)$. Hence, with our choice of $\varepsilon$, we have for all $t \geq t_1 \vee t_2$

$$|\psi(\mu, w) - \psi(\mu_t, w_t)| = \left| \min_{\lambda \in B(\mu)} f(\mu, \lambda, w) - \min_{\lambda \in B(\mu)} f(\mu_t, \lambda, w_t) \right|$$

$$\leq \varepsilon \left| \min_{\lambda \in B(\mu)} f(\mu, \lambda, w) \right|$$

$$\leq \varepsilon |\psi(\mu, w)|.$$

This concludes the proof of the continuity of $\psi$.

Now, we know that $w \mapsto \psi(\mu, w)$ is continuous on $\Lambda$, and by compactness of the simplex, the maximum is attained at some $w_\mu^\star \in \Lambda$. Furthermore, since $\mathcal{A}$ spans $\mathbb{R}^d$, we may construct an allocation $\tilde{w}$ such that $\sum_{a \in \mathcal{A}} \tilde{w}_a a a^\top$ is a positive definite matrix. In addition, by construction of $B(\lambda)$, there exists some $M > 0$ such that for all $\lambda \in B(\mu)$ we have $\|\mu - \lambda\| > M$, which implies that $\psi(\mu, \tilde{w}) \geq M^2 \lambda_{\min} \left( \sum_{a \in \mathcal{A}} \tilde{w}_a a a^\top \right) > 0$. On the other for any allocation $w \in \Lambda$ such that $\sum_{a \in \mathcal{A}} w_a a a^\top$ is rank deficient, we may find a $\lambda \in B(\mu)$ where $\lambda - \mu$ is in the null space of $\sum_{a \in \mathcal{A}} w_a a a^\top$. Therefore, $\sum_{a \in \mathcal{A}} (w_\mu^\star)_a a a^\top$ is invertible $\qquad \square$

## B.2  Proof of Lemma 2

The lemma is a direct consequence of the maximum theorem (a.k.a. Berge's theorem) [22] and only requires that $\psi$ is continuous in $(\mu, w) \in \mathbb{R}^d \times \Lambda$, that $\Lambda$ is compact, convex and non-empty, and that $\psi$ is concave in $w$ for each $\mu' \in \mathbb{R}^d$ in an open neighberhood of $\mu$. These requirements hold naturally in our setting: (i) by Lemma 1, we have for all $\mu \in \mathbb{R}^d$ such that $a_\mu^\star$ is unique and for any $w \in \Lambda$, $\psi$ is continuous in $(\mu, w)$; (ii) $\Lambda$ is a non-empty, compact and convex set; (iii) for all $\mu \in \mathbb{R}^d$, $w \mapsto \psi(\mu, w)$ is concave as it can be expressed as the infimum of linear functions in $w$. Therefore, the maximum theorem applies and we obtain the desired results. $\qquad \square$

# C Least Squares Estimator

In this appendix, we present concentration bounds and convergence statements on the least squares estimator. We may recall that the least squares estimation error $\hat{\mu}_t - \mu$ can be expressed conveniently in the following form[2]: $\hat{\mu}_t - \mu = (\sum_{s=1}^{t} a_s a_s^\top)^{-1}(\sum_{s=1}^{t} a_s \eta_s)$. To make notations less cluttered, we prefer to express our derivations in matrix form where we define the covariates matrix $A_t = [a_1 \ \dots \ a_t]^\top$ and noise vector $E_t = [\eta_1 \ \dots \ \eta_t]^\top$. We may then write $\hat{\mu}_t - \mu = (A_t^\top A_t)^{-1}(A_t^\top E_t)$. Furthermore, we will reapeatedly use the following decoposition

$$\|\hat{\mu}_t - \mu\| = \|(A_t^\top A_t)^{-1}(A_t^\top E_t)\| \leq \|A_t^\top E_t\|_{(A_t^\top A_t)^{-1}} \|(A_t^\top A_t)^{-1/2}\| \tag{13}$$

where we have $\|x\|_A = \sqrt{x^\top A x}$ for some semi-definite positive matrix $A$. The above inequality follows from Cauchy-Schwarz inequality. We also observe that when $A_t^\top A_t$ is invertible, we have $\|(A_t^\top A_t)^{-1/2}\| = \lambda_{\min}(A_t^\top A_t)^{-1/2}$.

## C.1 Self-Normalized processes

We first present convenient tools from the theory of self-normalized processes [23], namely the deviation bounds established by Abbasi-Yadkouri et al. in [9].

**Proposition 4** (Theorem 1. in [9]). Let $(\mathcal{F}_t)_{t \geq 0}$ be a filtration. Let $\{\eta_t\}_{t \geq 1}$ be a real-valued stochastic process such that for all $t \geq 1$, $\eta_t$ is $\mathcal{F}_{t-1}$-measurable and satisfies with some postive $\sigma$, the conditional $\sigma$-sub-gaussian condition: $\mathbb{E}\left[\exp(x\eta_t)|\mathcal{F}_{t-1}\right] \leq \exp\left(-x^2\sigma^2/2\right)$, for all $x \in \mathbb{R}$. Let $(a_t)_{t \geq 1}$ be an $\mathbb{R}^d$-valued stochastic process adapted to $\{\mathcal{F}_t\}_{t \geq 0}$. Furthermore, let $V$ be a positive definite matrix. Then for all $\delta \in (0,1)$ we have

$$\mathbb{P}\left(\left\|A_t^\top E_t\right\|_{(A_t^\top A_t + V)^{-1}}^2 \leq 2\sigma^2 \log\left(\det\left((A_t^\top A_t + V)V^{-1}\right)^{\frac{1}{2}}/\delta\right)\right) \geq 1 - \delta.$$

The following result is a stronger version of Proposition 4 and in fact is behind its proof.

**Proposition 5** (Lemma 9. in [9]). With the same assumptions as in the above proposition. Let $\tau$ be any stopping time with respect to the filtration $(\mathcal{F})_{t \geq 1}$. Then, for $\delta > 0$, we have

$$\mathbb{P}\left(\left\|A_\tau^\top E_\tau\right\|_{(A_\tau^\top A_\tau + V)^{-1}}^2 \leq 2\sigma^2 \log\left(\det\left((A_\tau^\top A_\tau + V)V^{-1}\right)^{\frac{1}{2}}/\delta\right)\right) \geq 1 - \delta.$$

## C.2 Proof of Lemma 3

Lemma 3 shows that the convergence rate of the least squares estimator is dictated by the growth rate of the smallest eigenvalue of the covariates matrx $A_t^\top A_t$. Parts of our proof technique are inspired by recent developments in learning dynamical systems [24].

*Proof.* Define the event

$$\mathcal{E} = \left\{\exists c > 0, \exists t_0 \geq 0, \forall t \geq t_0, \quad \frac{1}{t^\alpha}\lambda_{\min}(A_t^\top A_t) > c\right\}.$$

By assumption, $\mathcal{E}$ holds with probability 1. Note that the $t_0, c$ may be random here. It also holds on the event $\mathcal{E}$ that for all $t \geq t_0$ we have $2A_t^\top A_t \succ A_t^\top A_t + ct^\alpha$ which implies that $2(A_t^\top A_t + ct^\alpha)^{-1} \succ (A_t^\top A_t)^{-1}$. This means that on the event $\mathcal{E}$, for all $t \geq t_0$, we have $\|A_t^\top E_t\|_{(A_s^\top A_s)^{-1}}^2 < 2\|A_t^\top E_t\|_{(A_s A_s^\top + ct^\alpha)^{-1}}^2$. Then, using the decomposition (13) we obtain

$$\|\hat{\mu}_t - \mu\| < \frac{\sqrt{2}\|A_t^\top E_t\|_{(A_t^\top A_t + ct^\alpha)^{-1}}}{\lambda_{\min}\left(A_t^\top A_t\right)^{1/2}} < \frac{\sqrt{2}}{\sqrt{c}t^{\alpha/2}}\|A_t^\top E_t\|_{(A_t^\top A_t + ct^\alpha)^{-1}}. \tag{14}$$

We will show that $\|A_t^\top E_t\|_{(A_t^\top A_t + ct^\alpha)^{-1}} = o(t^\beta)$ a.s. for all $\beta > 0$. This will ensure immediately with the upper bound (14) that $\|\hat{\mu}_t - \mu\| = o(t^{-\beta})$ a.s. for all $\beta \in (0, \alpha/2)$. By Proposition 4, it holds for all $\beta > 0$ and $t \geq 0$

$$\mathbb{P}\left(\frac{1}{t^\beta}\|A_t^\top E_t\|_{(A_t^\top A_t + ct^\alpha)^{-1}} > \frac{\sigma}{t^\beta}\left(2\log\left(\det\left((A_t^\top A_t + ct^\alpha I_d)(ct^\alpha I_d)^{-1}\right)^{\frac{1}{2}}/\delta\right)\right)^{1/2}\right) \leq \delta.$$

Since $\mathcal{A}$ is finite, we may upper bound $\det\left((A_t^\top A_t + ct^\alpha I_d)(ct^\alpha I_d)^{-1}\right) \leq (L^2 t^{1-\alpha}/c + 1)^d$ where $L = \max_{a\in\mathcal{A}} \|a\|$ and deduce that

$$\mathbb{P}\left(\frac{1}{t^\beta}\|A_t^\top E_t\|_{(A_t^\top A_t + ct^\alpha)^{-1}} > \frac{\sigma}{t^\beta}\left(2\log\left(L^d t^{\frac{(1-\alpha)d}{2}}/c^{\frac{d}{2}}\delta\right)\right)^{1/2}\right) \leq \delta,$$

which we may rewrite after substitution as

$$\mathbb{P}\left(\frac{1}{t^\beta}\|A_t^\top E_t\|_{(A_t^\top A_t + ct^\alpha)^{-1}} > \varepsilon\right) \leq \frac{L^d}{c^{\frac{d}{2}}}t^{\frac{(1-\alpha)d}{2}}\exp\left(-\frac{\varepsilon^2 t^{2\beta}}{2\sigma^2}\right).$$

For all $\varepsilon > 0$, since $\sum_{t=1}^\infty t^{\frac{(1-\alpha)d}{2}}\exp(-\frac{\varepsilon^2 t^{2\beta}}{2\sigma^2}) < \infty$, we have

$$\sum_{t=1}^\infty \mathbb{P}\left(\frac{1}{t^\beta}\|A_t^\top E_t\|_{(A_t^\top A_t + ct^\alpha)^{-1}} > \varepsilon\right) < \infty.$$

Thus, by the first Borell-Cantelli lemma, we have for all $\varepsilon > 0$

$$\mathbb{P}\left(\left\{\frac{1}{t^\beta}\|A_t^\top E_t\|_{(A_t^\top A_t + ct^\alpha)^{-1}} > \varepsilon\right\} i.o.\right) = 0.$$

Thus, we have proved that $\frac{1}{t^\beta}\|A_t^\top E_t\|_{(A_t^\top A_t + ct^\alpha)^{-1}} \xrightarrow[t\to\infty]{} 0$ a.s.. $\qquad\square$

## C.3 Proof of Lemma 4

The proof of Lemma 4 is very similar to that of Lemma 3, but in order to obtain a non-asymptotic concentration bound, a stronger condition is needed, namely a non-asymptotic lower bound for the rate of growth of the smallest eigenvalue of the covariates matrix $A_t^\top A_t$.

*Proof.* We have by assumption that there are $c > 0$ and $t_0 \geq 0$ such that for all $t \geq t_0$, the event

$$\mathcal{E} = \left\{\lambda_{\min}\left(A_t^\top A_t\right) > ct^\alpha\right\}$$

holds with probability 1. We can now carry the same derivation as in the proof of Lemma C.2 with the distinction that $c, t_0$ are deterministic and conclude that for all $\varepsilon > 0$, and $t \geq t_0$, we have

$$\mathbb{P}\left(\frac{\sqrt{2}}{\sqrt{c}t^\beta}\|A_t^\top E_t\|_{(A_t^\top A_t + ct^\alpha)^{-1}} > \varepsilon\right) \leq \frac{L^d}{c^{\frac{d}{2}}}t^{\frac{(1-\alpha)d}{2}}\exp\left(-\frac{c\varepsilon^2 t^{2\beta}}{4\sigma^2}\right),$$

with the choice of $\beta = \alpha/2$ and using the upper bound (14) which can be shown similarly under the event $\mathcal{E}$, we have for all $\varepsilon > 0$, and $t \geq t_0$ that

$$\mathbb{P}\left(\|\hat{\mu}_t - \mu\| > \varepsilon\right) \leq (c^{-1/2}L)^d t^{\frac{(1-\alpha)d}{2}}\exp\left(-\frac{\varepsilon^2 t^\alpha}{2\sigma^2}\right).$$

$\qquad\square$

# D  Stopping rule

The derivation of our stopping rule is inspired by that of Garivier and Kaufmann [18] for the MAB setting and relies on the classical generalized log-likelihood ratio (GLLR) test. The main distinction is that in the linear bandit setting, sampling an arm may provide additional statistical information about other arms, therefore one has to consider the full history of observations and sampled arms when comparing arms in the GLLR. We define our GLLR accordingly.

Furthermore, because of the linear structure, we are able to derive an exploration threshold which does not depend on the number of arms $K$, but only on the ambient dimension $d$. Our choice of threshold relies on the deviation bound presented in Proposition 5 (see Lemma 9 in [9]). But most importantly, to circumvent a naive union bound over the set of arms $\mathcal{A}$, we analyze the stopping time by leveraging the GLLR formulation (see Lemma 7) under the event of failure (failure to output the best arm). The stopping rules derived by Soare et al. [12] follow directly from the deviation bound in [9], rather than from the GLLR and consequently, they cannot avoid the dependency on $K$ even for the oracle stopping rule. Most existing algorithms in the literature are phase-based and rely on elimination criteria to stop [16, 15, 14]. In these algorithms, the phase transition rules and elimination criteria depend in a way or another on the number of arms $K$.

## D.1  Proof of Lemma 7

Here, we show that the generalized log-likelihood ratio can be expressed in a closed form, one that resembles the expression of $\psi$ used in the lower bound.

Let us first recall that, under the gaussian noise assumption, the density function of the sample path $r_1, a_1, \ldots, r_t, a_t$ is

$$f(r_1, a_1, \ldots, r_t, a_t) \propto \exp\left(-\frac{1}{2}\sum_{s=1}^{t}(r_s - \mu^\top a_s)^2\right).$$

Observe that the maximization problem $\max_{\{\mu : \mu^\top(a-b) \geq -\varepsilon\}} f_\mu(r_t, a_t, \ldots, r_1, a_1)$ is, by monotonicity of the exponential, equivalent to

$$\min_\mu \quad \frac{1}{2}\sum_{s=1}^{t}(r_s - \mu^\top a_s)^2$$
$$\text{s.t.} \quad \mu^\top(a-b) \geq -\varepsilon,$$

which is a convex program. The optimality conditions give us

$$\lambda \geq 0,$$
$$\lambda(\varepsilon + \mu^\top(a-b)) = 0,$$
$$-\varepsilon - \mu^\top(a-b) \leq 0,$$
$$\left(\sum_{s=1}^{t} a_s a_s^\top\right)\mu - \sum_{s=1}^{t} a_s r_s + \lambda(a-b) = 0,$$

where $\lambda$ is the Lagrange multiplier associated with the inequality constraint of the problem. Under the assumption that $\sum_{s=1}^{t} a_s a_s^\top$ is invertible, we introduce the least squares estimator $\hat{\mu}_t = \left(\sum_{s=1}^{t} a_s a_s^\top\right)^{-1}\left(\sum_{s=1}^{t} a_s r_s\right)$. Then from optimality conditions, it follows that

$$\mu_1^* = \begin{cases} \hat{\mu}_t & \text{if} \quad \hat{\mu}_t^\top(a-b) \geq -\varepsilon, \\ \hat{\mu}_t + (-\varepsilon - \hat{\mu}_t^\top(a-b))\dfrac{\left(\sum_{s=1}^{t} a_s a_s^\top\right)^{-1}(a-b)}{(a-b)^\top\left(\sum_{s=1}^{t} a_s a_s^\top\right)^{-1}(a-b)} & \text{otherwise.} \end{cases} \tag{15}$$

Similarly the solution to the maximization problem $\max_{\{\mu : \langle\mu, a-b\rangle \leq \varepsilon\}} f_\mu(r_t, a_t, \ldots, r_1, a_1)$ is

$$\mu_2^* = \begin{cases} \hat{\mu}_t & \text{if} \quad \hat{\mu}_t^\top(a-b) \leq -\varepsilon, \\ \hat{\mu}_t + (-\varepsilon - \hat{\mu}_t^\top(a-b))\dfrac{\left(\sum_{s=1}^{t} a_s a_s^\top\right)^{-1}(a-b)}{(a-b)^\top\left(\sum_{s=1}^{t} a_s a_s^\top\right)^{-1}(a-b)} & \text{otherwise.} \end{cases} \tag{16}$$

Hence, the generalized log likelihood ratio can be expressed as

$$Z_{a,b,\varepsilon}(t) = \frac{1}{2}(\mu_1^* - \mu_2^*)^\top \left(\sum_{s=1}^t a_s a_s^\top\right)(2\mu_t - \mu_1^* - \mu_2^*)$$

$$= \text{sign}(\mu_t^\top(a-b) + \varepsilon)\frac{(\hat{\mu}_t^\top(a-b) + \varepsilon)^2}{2(a-b)^\top\left(\sum_{s=1}^t a_s a_s^\top\right)^{-1}(a-b)}.$$

□

The following corollary is an immediate consequence of Lemma 7. Let us recall that $Z_{a,b}(t) = Z_{a,b,0}(t)$.

**Corollary 1.** Let $t \geq 0$, and assume that $\sum_{s=1}^t a_s a_s^\top \succ 0$. Then for all $\hat{a}_t \in \arg\max_{a \in \mathcal{A}} \hat{\mu}_t^\top a$, it holds

$$Z(t) = \max_{a \in \mathcal{A}} \min_{b \in \mathcal{A}\setminus\{a\}} Z_{a,b}(t) = \min_{b \in \mathcal{A}\setminus\{\hat{a}_t\}} Z_{\hat{a}_t,b}(t). \tag{17}$$

*Proof.* Under the assumption that $\sum_{s=1}^t a_s a_s^\top \succ 0$, by Lemma 7, the sign of $Z_{a,b}(t)$ is that of $\hat{\mu}_t^\top(a-b)$. Additionally, since $\hat{a}_t \in \arg\max_{a \in \mathcal{A}} \hat{\mu}_t^\top a$, it holds for all $b \in \mathcal{A}\setminus\{\hat{a}_t\}$ that $\hat{\mu}_t^\top(\hat{a}_\tau - b) \geq 0$. Hence it immediately follows that $Z_{a,b}(t) \geq 0$ if and only if $a \in \arg\max_{a \in \mathcal{A}} \hat{\mu}_t^\top a$. Furthermore, if $\hat{a}_t$ is not unique, then we may find $b \in \arg\max \hat{\mu}_t^\top b$ such that $\hat{a}_t \neq b$, and then by Lemma 7 obtain $Z_{\hat{a}_t,b}(t) = 0$. Hence, we conclude that regardless of whether $\hat{a}_t$ is unique or not, $Z(t) = \min_{b \in \mathcal{A}\setminus\{\hat{a}_t\}} Z_{\hat{a}_t,b}(t)$. □

### D.2 Proof of Proposition 2

Let us consider the events

$$\mathcal{E}_1 = \{\tau < \infty\} = \left\{\exists t \in \mathbb{N}^* : \max_{a \in \mathcal{A}} \min_{b \in \mathcal{A}\setminus\{a\}} Z_{a,b}(t) > \beta(\delta, t) \text{ and } \sum_{s=1}^t a_s a_s^\top \succeq cI_d\right\},$$

$$\mathcal{E}_2 = \{\mu^\top(a_\mu^* - \hat{a}_\tau) > 0\}.$$

Now note that if there exists $t \in \mathbb{N}^*$ such that $\sum_{s=1}^t a_s a_s \succeq cI_d$ and $\mu^\top(a_\mu^* - \hat{a}_t) > 0$ then $\hat{a}_t \neq a_\mu^*$. Additionally, from Corollary 1, we know that under $\mathcal{E}_1$, that for all $t \geq 1$, it holds that $Z(t) = \min_{b \in \mathcal{A}\setminus\{\hat{a}_t\}} Z_{\hat{a}_t,b}(t)$. Therefore, we have

$$\mathcal{E}_1 \cap \mathcal{E}_2 = \left\{\exists t \in \mathbb{N}^* : Z(t) > \beta(\delta, t) \text{ and } \sum_{s=1}^t a_s a_s^\top \succeq cI_d \text{ and } \mu^\top(a_\mu^\star - \hat{a}_t) > 0\right\}$$

$$= \left\{\exists t \in \mathbb{N}^* : \min_{b \in \mathcal{A}\setminus\{\hat{a}_t\}} Z_{\hat{a}_t,b}(t) > \beta(\delta, t) \text{ and } \sum_{s=1}^t a_s a_s^\top \succeq cI_d \text{ and } \mu^\top(a_\mu^\star - \hat{a}_t) > 0\right\}$$

$$\subseteq \left\{\exists t \in \mathbb{N}^* : Z_{\hat{a}_t,a_\mu^\star}(t) > \beta(\delta, t) \text{ and } \sum_{s=1}^t a_s a_s^\top \succeq cI_d \text{ and } \mu^\top(a_\mu^\star - \hat{a}_t) > 0\right\}.$$

Since under the event $\mathcal{E}_1 \cap \mathcal{E}_2$ and by definition of $\hat{a}_t$, we have $\hat{\mu}_t^\top(\hat{a}_t - a_\mu^\star) \geq 0$, and $\mu^\top(a_\mu^\star - \hat{a}_t) > 0$. In view of (15), it follows that

$$\max_{\{\mu':(\mu')^\top(\hat{a}_t - a_\mu^\star)\geq 0\}} f_{\mu'}(r_t, a_t, \ldots, r_1, a_1) = f_{\hat{\mu}_t}(r_t, a_t, \ldots, r_1, a_1),$$

$$\max_{\{\mu':(\mu')^\top(\hat{a}_t - a_\mu^\star)\leq 0\}} f_{\mu'}(r_t, a_t, \ldots, r_1, a_1) \geq f_\mu(r_t, a_t, \ldots, r_1, a_1).$$

Thus under $\mathcal{E}_1 \cap \mathcal{E}_2$ it holds that

$$Z_{\hat{a}_t, a_\mu^*}(t) = \log \left( \frac{\max_{\mu' : (\mu')^\top (\hat{a}_t - a_\mu^\star) \geq 0} f_{\mu'}(r_t, a_t, \ldots, r_1, a_1)}{\max_{\mu' : (\mu')^\top (\hat{a}_t - a_\mu^\star) \leq 0} f_{\mu'}(r_t, a_t, \ldots, r_1, a_1)} \right)$$

$$\leq \log \left( \frac{f_{\hat{\mu}_t}(r_t, a_t, \ldots, r_1, a_1)}{f_\mu(r_t, a_t, \ldots, r_1, a_1)} \right)$$

$$= \frac{1}{2} (\hat{\mu}_t - \mu)^\top \left( \sum_{s=1}^t a_s a_s^\top \right) (\hat{\mu}_t - \mu)$$

$$= \frac{1}{2} \|\mu - \hat{\mu}_t\|_{\sum_{s=1}^t a_s a_s^\top}^2,$$

which further implies that

$$\mathcal{E}_1 \cap \mathcal{E}_2 \subseteq \left\{ \exists t \in \mathbb{N}^* : \quad \frac{1}{2} \|\mu - \hat{\mu}_t\|_{\sum_{s=1}^t a_s a_s^\top}^2 \geq \beta(\delta, t) \text{ and } \sum_{s=1}^t a_s a_s^\top \succeq c I_d \right.$$

$$\left. \text{and } \mu^\top (a_\mu^* - \hat{a}_t) > 0 \right\}$$

$$\subseteq \left\{ \exists t \in \mathbb{N}^* : \quad \frac{1}{2} \|\mu - \mu_t\|_{\sum_{s=1}^t a_s a_s^\top}^2 \geq \beta(\delta, t) \text{ and } \sum_{s=1}^t a_s a_s^\top \succeq c I_d \right\}.$$

We note that when $\sum_{s=1}^t a_s a_s^\top \succeq c I_d$, then for all $\rho > 0$, $(1+\rho) \sum_{s=1}^t a_s a_s^\top \succeq \sum_{s=1}^t a_s a_s^\top + \rho c I_d$, which means that $(1+\rho)(\sum_{s=1}^t a_s a_s^\top + \rho c I_d)^{-1} \succeq (\sum_{s=1}^t a_s a_s^\top)^{-1}$. Thus, we may have

$$\|\hat{\mu}_t - \mu\|^2 = \left\| \sum_{s=1}^t a_s \eta_s \right\|_{(\sum_{s=1}^t a_s a_s^\top)^{-1}}^2 \leq (1+\rho) \left\| \sum_{s=1}^t a_s \eta_s \right\|_{(\sum_{s=1}^t a_s a_s^\top + \rho c I_d)^{-1}}^2.$$

This leads to

$$\mathcal{E}_1 \cap \mathcal{E}_2 \subseteq \left\{ \exists t \in \mathbb{N}^* : \quad \frac{1}{2}(1+\rho) \left\| \sum_{s=1}^t a_s \eta_s \right\|_{(\sum_{s=1}^t a_s a_s^\top + \rho c I_d)^{-1}}^2 \geq \beta(\delta, t) \right\},$$

and with the choice

$$\beta(\delta, t) = (1+\rho)\sigma^2 \log \left( \frac{\det((\rho c)^{-1} \sum_{s=1}^t a_s a_s^\top + I_d)^{1/2}}{\delta} \right),$$

we write

$$\mathcal{E}_1 \cap \mathcal{E}_2 \subseteq$$
$$\left\{ \exists t \in \mathbb{N}^* : \frac{1}{2} \left\| \sum_{s=1}^t a_s \eta_s \right\|_{(\sum_{s=1}^t a_s a_s^\top + \rho c I_d)^{-1}}^2 > \sigma^2 \log \left( \frac{\det((\rho c)^{-1} \sum_{s=1}^t a_s a_s^\top + I_d)^{1/2}}{\delta} \right) \right\}.$$

Finally, it follows immediately from Proposition 5 that

$$\mathbb{P}\left( \tau < \infty, \mu^\top (a_\mu^\star - \hat{a}_\tau) > 0 \right) = \mathbb{P}(\mathcal{E}_1 \cap \mathcal{E}_2) \leq \delta.$$

$\square$

Proposition 2 does not yet guarantee that we have a $\delta$-PAC startegy. However, a sufficient condition for any strategy with the proposed decision rule and stopping rule to be $\delta$-PAC, is to simply ensure that $\mathbb{P}(\tau < \infty) = 1$. This condition will have to be satisfied by our sampling rule.

**Corollary 2** ($\delta$-PAC guarantee). For any strategy using the proposed decision rule and stopping rule and such that $\mathbb{P}(\tau < \infty)$, it is guaranteed that $\mathbb{P}(\mu^\top (a_\mu^\star - \hat{a}_t) > 0) \leq \delta$.

# E  Sampling rule

Our sampling rule as described in Section 3.5 is based on tracking a sequence of allocations that provably approaches the set of optimal allocations. This set of optimal allocations $C^\star(\mu)$ that is not necessarily a singleton as in the multi-armed bandit setting [20]. This makes the analysis extremely challenging. However by crucially leveraging the geometric properties of this set and the continuity properties of $\psi$ and $C^\star(\mu)$ we are able to prove that tracking is possible.

Additionally, we choose arms from the support (set of non zero elements) of the average allocations up to the current round. This is motivated by the fact that when $K$ is exceedingly large in comparison with the dimension $d$, it is possible to represent any matrix $A$ in the convex hull $\mathrm{conv}(\{aa^\top : a \in \mathcal{A}\})$ by an allocation $w$ with support of at most $O(d^2)$ such that $A = \sum_{a \in \mathcal{A}} w_a aa^\top$. This observation was made by Soare et el. [12] and follows from Caratheorody's Theorem. A consequence of this sampling strategy is reflected in Lemma 6.

One further novel part of the analysis is the introduction of *laziness*, the idea that the algorithm does not need to perform a computationally demanding task at every round. In the linear bandit setting this computationally demanding task is the optimization problem $\max_{w \in \Lambda} \psi(\hat{\mu}_t, w)$. Existing algorithms in the literature resort to phase-based schemes such us gap elimination in order to attain efficiency. However these schemes often fail to fully stitch the statistical information between phases. This can be seen in the least squares constructions of the algorithms $\mathcal{XY}$-adaptive [12], ALBA [15], RAGE [16] where the samples from previous phases are discarded. Our tracking rule allows for a natural flow of information between rounds regardless of the laziness of the algorithm. This is shown by Proposition 1.

We shall now prove Proposition 1 and all the related lemmas. Lemma 5 shows that we have sufficient exploration. Lemma 6 is the crucial step in our analysis here. It's a tracking lemma that formalizes the idea that we may track a sequence that converges to a set $C$ rather than a point. The proof requires the convexity of the set $C$. In the main analysis of the sampling rule $C$ is replaced by $C^\star(\mu)$.

## E.1  Proof of Lemma 5

The idea of the proof is to show that if at some time $t_0 + 1$, the condition $\lambda_{\min}(\sum_{s=1}^{t} a_s a_s^\top) > f(t)$ is violated, then the number of rounds needed to satisfy the condition again cannot exceed $d$ rounds.

First, we note that $d = \inf\{t \geq 1 : \lambda_{\min}(\sum_{s=1}^{t} a_s a_s^\top) \geq f(t)\}$. Indeed, we have by construction that for all $t < d$, $\lambda_{\min}(\sum_{s=1}^{t} a_s a_s^\top) = 0$ and $\lambda_{\min}\left(\sum_{s=1}^{d} a_s a_s^\top\right) = \lambda_{\min}\left(\sum_{a \in \mathcal{A}_0} aa^\top\right) = f(d)$. Now,

if there exists $t_0 \geq d$, such that $\lambda_{\min}\left(\sum_{s=1}^{t_0} a_s a_s^\top\right) \geq f(t_0)$ and $\lambda_{\min}\left(\sum_{s=1}^{t_0+1} a_s a_s^\top\right) < f(t_0 + 1)$, then we may define $t_1 = \inf\left\{t > t_0 : \lambda_{\min}\left(\sum_{s=1}^{t} a_s a_s^\top\right) \geq f(t)\right\}$. Let us observe that for all $t_0 \leq t \leq t_1$, we have

$$\lambda_{\min}\left(\sum_{s=1}^{t} a_s a_s^\top\right) \geq \lambda_{\min}\left(\sum_{s=1}^{t_0} a_s a_s^\top\right) \geq f(t_0).$$

Note that if $t_1 \geq t_0 + d + 1$, then, by construction, we have

$$
\lambda_{\min}\left(\sum_{s=1}^{t_1} a_s a_s^\top\right) \geq \lambda_{\min}\left(\sum_{s=1}^{t_0+d+1} a_s a_s^\top\right)
$$

$$
\geq \lambda_{\min}\left(\sum_{s=1}^{t_0+1} a_s a_s^\top + \sum_{a \in \mathcal{A}_0} a a^\top\right)
$$

$$
\geq \lambda_{\min}\left(\sum_{s=1}^{t_0+1} a_s a_s^\top\right) + \lambda_{\min}\left(\sum_{a \in \mathcal{A}_0} a a^\top\right)
$$

$$
= \lambda_{\min}\left(\sum_{s=1}^{t_0+1} a_s a_s^\top\right) + c_{\mathcal{A}_0}\sqrt{d}
$$

$$
\geq \lambda_{\min}\left(\sum_{s=1}^{t_0} a_s a_s^\top\right) + c_{\mathcal{A}_0}\sqrt{d}
$$

$$
\geq f(t_0) + c_{\mathcal{A}_0}\sqrt{d}.
$$

However, we have

$$
t_0 \geq \frac{1}{4}\left(d + \frac{1}{d} + 2\right) \implies \sqrt{t_0 + d + 1} + \sqrt{t_0} \geq \sqrt{d} + \frac{1}{\sqrt{d}} \implies f(t_0) + c_{\mathcal{A}_0}\sqrt{d} \geq f(t_0 + d + 1).
$$

Therefore, if $t_0 \geq \frac{1}{4}\left(d + \frac{1}{d} + 2\right)$, then it holds that $t_1 \leq t_0 + d + 1$. In other words, we have shown that for all $t \geq \frac{1}{4}\left(d + \frac{1}{d} + 2\right) + d + 1$, we have

$$
\lambda_{\min}\left(\sum_{s=1}^{t} a_s a_s^\top\right) \geq f(t - d - 1).
$$

$\square$

## E.2  Proof of Lemma 6

Our proof for the tracking lemma is inspired by that of D-tracking for linear bandits by Garivier and Kaufmann [18]. We follow similar steps but there are crucial differences. The main one lies in the fact that we have a sequence that converges to a set $C$ rather than to a unique point. The convexity of $C$ is a crucial point in our analysis as it allows to show that tracking the average of this converging sequence will eventually allow our empirical allocation to be sufficiently close to the set $C$. Intuitively, the average is a stable point to track. Furthermore, we also highlight the fact that the sparsity of the average allocations $\sum_{s=1}^{t} w(s)/t$ is reflected in the error by which $(N_a(t))_{a \in \mathcal{A}}$ approaches the set $C$. This is due to the nature of our sampling rule as shall be proven.

*Proof.* For all $t \geq 1$ denote

$$
\overline{w}(t) = \frac{1}{t}\sum_{s=1}^{t} w(s).
$$

Since $C$ is non-empty and compact, we may define

$$
\hat{w}(t) = \arg\min_{w \in C} d_\infty(\overline{w}(t), w).
$$

Note that by convexity of $C$, there exists $t_0' \geq t_0$ such that $\forall t \geq t_0'$, $d_\infty((N_a(t)/t)_{a \in \mathcal{A}}, C) \leq d_\infty((N_a(t)/t)_{a \in \mathcal{A}}, \hat{w}(t))$ and $d_\infty(\overline{w}(t), \hat{w}(t)) \leq 2\varepsilon$.

To see that, let us define for all $t \geq 1$, $v(t) = \arg\min_{w \in C} d_\infty(w, w(t))$, and observe that for all $a \in \mathcal{A}$, we have

$$
\left|\frac{1}{t}\sum_{s=1}^{t} w_a(s) - \frac{1}{t}\sum_{s=1}^{t} v_a(s)\right| \leq \frac{1}{t}\sum_{s=1}^{t_0} |w_a(s) - v_a(s)| + \frac{1}{t}\sum_{s=t_0+1}^{t} |w_a(s) - v_a(s)|
$$

$$
\leq \frac{t_0}{t} + \frac{t - t_0}{t}\varepsilon.
$$

Thus if $t \geq t_0' = \frac{t_0}{\varepsilon}$, then $d_\infty(\overline{w}(t), \frac{1}{t}v(t)) \leq 2\varepsilon$. Finally since $\frac{1}{t}\sum_{s=1}^{t} v(s) \in C$ (by convexity of $C$), it follows that

$$\forall t \geq t_0' \qquad d_\infty(\overline{w}(t), \hat{w}(t)) \leq d_\infty\left(\overline{w}(t), \frac{1}{t}\sum_{s=1}^{t} v(s)\right) \leq 2\varepsilon.$$

We further define for all $t \geq 1$, $\varepsilon_{a,t} = N_a(t) - t\hat{w}_a(t)$. The main step of the proof is to show that there exists $t_0'' \geq t_0'$ such that for all $t \geq t_0''$, for all $a \in \mathcal{A}$ we have

$$\{a_{t+1} = a\} \subseteq \mathcal{E}_1(t) \cup \mathcal{E}_2(t) \subseteq \{\varepsilon_{a,t} \leq 6t\varepsilon\},$$

where

$$\mathcal{E}_1(t) = \left\{ a = \underset{a \in \mathrm{supp}(w_t)}{\arg\min}\left(N_a(t) - t\overline{w}_a(t)\right) \right\},$$

$$\mathcal{E}_2(t) = \left\{ \lambda_{\min}\left(\sum_{s=1}^{t} a_s a_s^\top\right) < f(t) \quad and \quad a = \mathcal{A}_0(i_t) \right\}.$$

The first inclusion is immediate by construction. Now let $t \geq t_0$, we have:

*(Case 1)* If $\{a_{t+1} = a\} \subseteq \mathcal{E}_1(t)$, then we have

$$
\begin{aligned}
\varepsilon_{a,t} &= N_a(t) - t\hat{w}_a(t) \\
&= N_a(t) - t\overline{w}_a(t) + t\overline{w}_a(t) - t\hat{w}_a(t) \\
&\leq N_a(t) - t\overline{w}_a(t) + t\varepsilon && \text{(since } d_\infty(\hat{w}(t), \overline{w}(t)) \leq \varepsilon) \\
&\leq \underset{a \in \mathrm{supp}(\overline{w}(t))}{\min} N_a(t) - t\overline{w}_a(t) + t\varepsilon && \text{(since } \mathcal{E}_1(t) \text{ holds)} \\
&\leq 2t\varepsilon,
\end{aligned}
$$

where the last inequality holds because

$$\sum_{a \in \mathrm{supp}(\overline{w}(t))} N_a(t) - t\overline{w}_a(t) = - \sum_{a \in \mathcal{A} \backslash \mathrm{supp}(\overline{w}(t))} N_a(t) \leq 0$$

thus $\mathcal{E}_1(t) \subseteq \{\varepsilon_{a,t} \leq 2t\varepsilon\}$.

*(Case 2)* If $\{a_{t+1} = a\} \subseteq \mathcal{E}_2(t)$, then it must hold that $a \in \mathcal{A}_0$. Let us define for al $k \geq 1$

$$N_{a,1}(k) = \sum_{s=1}^{k} \mathbb{1}_{\left\{a_k=a \text{ and } \lambda_{\min}\left(\sum_{s=1}^{k-1} a_s a_s^\top\right) < f(k-1)\right\}},$$

$$N_{a,2}(k) = \sum_{s=1}^{k} \mathbb{1}_{\left\{a_k=a \text{ and } \lambda_{\min}\left(\sum_{s=1}^{k-1} a_s a_s^\top\right) \geq f(k-1)\right\}}.$$

Note that $N_a(k) = N_{a,1}(k) + N_{a,2}(k)$ and that $N_{a,1}(k) - 1 \leq \min_{a \in \mathcal{A}_0} N_{a,1}(k) \leq N_{a,1}(k)$. The latter property follows from the forced exploration sampling scheme. Now, since the event $\mathcal{E}_2(t)$ holds, we observe that

$$(N_{a,1}(t) - 1) \leq \min_{a \in \mathcal{A}_0} N_{a,1}(t)\lambda_{\min}\left(\sum_{a \in \mathcal{A}_0} aa^\top\right) \leq \lambda_{\min}\left(\sum_{s=1}^{t} a_s a_s^\top\right) < f(t)$$

and since $f(t) = \lambda_{\min}\left(\sum_{a \in \mathcal{A}_0} aa^\top\right)\frac{\sqrt{t}}{\sqrt{d}}$, we obtain

$$N_{a,1}(t) \leq \sqrt{t}/\sqrt{d} + 1.$$

Next, let $k \leq t$ be the largest integer such that $N_{a,2}(k) = N_{a,2}(k-1) + 1$. Note that at such $k$ the event $\mathcal{E}_1(k-1)$ must hold by definition of $N_{a,2}(k-1)$, and we have

$$N_{a,2}(t) = N_{a,2}(k) = N_{a,2}(k-1) + 1 \quad and \quad a_k = \underset{a \in \mathrm{supp}(w(k-1))}{\arg\min} N_a(k-1) - k\overline{w}_a(k-1).$$

Now we write

$$\varepsilon_{a,t} = N_{a,t} - t\hat{w}_a(t)$$
$$= N_{a,1}(t) + N_{a,2}(t) - t\hat{w}_a(t)$$
$$\le \sqrt{t}/\sqrt{d} + 1 + N_{a,2}(t) - t\hat{w}_a(t).$$

If $k - 1 \le t_0'$, then we have $N_{a,2}(k) \le t_0'$, otherwise since $\mathcal{E}_1(k-1)$ holds, we have

$$N_{a,2}(t) = 1 + N_{a,2}(k-1) - (k-1)\hat{w}_a(k-1) + (k-1)\hat{w}_a(k-1)$$
$$\le 1 + 2(k-1)\varepsilon + (k-1)\hat{w}_a(k-1).$$

Thus

$$\varepsilon_{a,t} \le \sqrt{t}/\sqrt{d} + 1 + \max\{t_0', 1 + 2(k-1)\varepsilon + (k-1)\hat{w}_a(k-1) - t\hat{w}_a(t)\},$$

and since

$$(k-1)\hat{w}_a(k-1) - t\hat{w}_a(t) = (k-1)\hat{w}_a(k-1) - (k-1)\overline{w}_a(k-1)$$
$$+ (k-1)\overline{w}_a(k-1) - t\hat{w}_a(t)$$
$$\le (k-1)\hat{w}_a(k-1) - (k-1)\overline{w}_a(k-1) + t\overline{w}_a(t) - t\hat{w}_a(t)$$
$$\le 2(k-1)\varepsilon + 2t\varepsilon$$
$$\le 4t\varepsilon,$$

it follows that

$$\varepsilon_{a,t} \le \sqrt{t}/\sqrt{d} + 1 + \max\{t_0', 1 + 6t\varepsilon\}.$$

We conclude that for $t \ge t_0'' = \max\left\{\frac{1}{\varepsilon}, \frac{1}{\varepsilon^2 d}, \frac{t_0'}{\varepsilon}\right\}$, it holds that

$$\varepsilon_{a,t} \le 9t\varepsilon$$

and consequently that $\mathcal{E}_2(t) \subseteq \{\varepsilon_{a,t} \le 9t\varepsilon\}$. So we have shown that for all $t \ge t_0''$, for all $a \in \mathcal{A}$, it holds that

$$\{a_{t+1} = a\} \subseteq \{\varepsilon_{a,t} \le 9t\varepsilon\}.$$

The remaining part of the proof is very similar to that of Lemma 17 in [18]. It can be immediately shown that for $t \ge t_0''$, one has

$$\varepsilon_{a,t} \le \max(\varepsilon_{a,t_0''}, 9t\varepsilon + 1) \le \max(t_0'', 9t\varepsilon + 1)$$

Furthermore, note that for all $t \ge 1$ we have $\mathrm{supp}(\overline{w}(t)) \subseteq \mathrm{supp}(\overline{w}(t+1))$ since for all $a \in \mathcal{A}$, we have $t\overline{w}_a(t) \le (t+1)\overline{w}_a(t+1)$. Therefore

$$\sum_{a \in \mathrm{supp}(\overline{w}(t)) \cup \mathcal{A}_0} \varepsilon_{a,t} = \sum_{a \in \mathcal{A} \backslash \mathrm{supp}(\overline{w}(t)) \cup \mathcal{A}_0} t\hat{w}_a(t) \ge 0.$$

Thus denoting $p_t = |\mathrm{supp}(\overline{w}(t))| \backslash \mathcal{A}_0$, we have

$$\forall a \in \mathrm{supp}(\overline{w}(t)) \cup \mathcal{A}_0, \quad \max(t_0'', 9t\varepsilon + 1) \ge \varepsilon_{a,t} \ge -(p_t + d - 1)\max(t_0'', 9t\varepsilon + 1),$$
$$\forall a \in \mathcal{A} \backslash \mathrm{supp}(\overline{w}(t)) \cup \mathcal{A}_0, \qquad 0 \ge \varepsilon_{a,t} \ge -t\varepsilon,$$

which implies that for all $t \ge t_0''$

$$\max_{a \in \mathcal{A}} |\varepsilon_{a,t}| \le (p_t + d - 1)\max(t_0'', 9t\varepsilon + 1) \le (p_t + d - 1)\max(t_0'', 10).$$

This finally implies that for $t_1 = \frac{1}{\varepsilon}\max\{t_0'', 10\}$, we have for all $t \ge t_1$,

$$d_\infty(x(t), C^*) \le d_\infty((N_a(t)/t)_{a \in \mathcal{A}}, \hat{w}(t)) = \max_{a \in \mathcal{A}} |N_a(t)/t - \hat{w}_a(t)| = \max_{a \in \mathcal{A}} \left|\frac{\varepsilon_{a,t}}{t}\right| \le (p_t + d - 1)\varepsilon.$$

More precisely, we have

$$t_1(\varepsilon) = \max\left\{\frac{1}{\varepsilon^2}, \frac{1}{\varepsilon^3 d}, \frac{t_0(\varepsilon)}{\varepsilon^3}, \frac{10}{\varepsilon}\right\}.$$

$\square$

### E.3  Proof of Proposition 1

Let $\varepsilon > 0$. First, by Lemma 2, there exists $\xi(\varepsilon) > 0$ such that for all $\mu'$ such that $\|\mu - \mu'\| < \xi(\varepsilon)$, it holds that $\max_{w \in C^\star(\mu')} d_\infty(w, C^\star(\mu)) < \varepsilon/2$.

By Lemma 5, we have a sufficient exploration. That is $\liminf_{t \to \infty} t^{-1/2} \lambda_{\min}(\sum_{s=1}^t a_s a_s^\top) > 0$. Thus, by Lemma 3, $\hat{\mu}_t$ converges almost surely to $\mu$ with a rate of order $o(t^{1/4})$. Consequently, there exists $t_0 \geq 0$ such that for all $t \geq t_0$, we have $\|\mu - \hat{\mu}_t\| \leq \xi(\varepsilon)$.

The lazy condition (7) states that there exists a sequence $(\ell(t))_{t \geq 1}$ of integers such that $\ell(1) = 1$, $\ell(t) \leq t$ and $\lim_{t \to \infty} \ell(t) = \infty$, and $\lim_{t \to \infty} \inf_{s \geq \ell(t)} d_\infty(w(t), C^\star(\hat{\mu}_s)) = 0$ a.s. This guarantees that there exists $t_1 \geq 1$, there exists a sequence $(h(t))_{t \geq 1}$ of integers such that for all $t \geq t_1$, we have $h(t) \geq \ell(t) \geq t$ and $d_\infty(w(t), C^\star(\hat{\mu}_{h(t)})) < \varepsilon/2$. Now for all $t \geq t_0 \vee t_1$, we have

$$d_\infty(w(t), C^\star(\mu)) \leq d_\infty(w(t), C^\star(\hat{\mu}_{h(t)})) + \max_{w \in C^\star(\hat{\mu}_{h(t)})} d_\infty(w, C^\star(\mu)) < \varepsilon.$$

We have shown that $d_\infty(w(t), C^\star(\mu)) \xrightarrow[t \to \infty]{} 0$ a.s. Next, we recall that by Lemma 2, $C^\star(\mu)$ is non empty, compact and convex. Thus, applying Lemma 6 yields immediately that $d_\infty((N_a(t)/t)_{a \in \mathcal{A}}, C^\star(\mu)) \xrightarrow[t \to \infty]{} 0$ a.s.. $\qquad\square$

# F  Sample complexity

We will use the following technical lemma which can be found for instance in [18].

**Lemma 8** (Lemma 18 [18]). *For any two constants $c_1, c_2 > 0$, and $c_2/c_1 > 1$ we have*

$$\inf\left\{t \in \mathbb{N}^* : \ c_1 t \geq \log(c_2 t)\right\} \leq \frac{1}{c_1}\left(\log\left(\frac{c_2 e}{c_1}\right) + \log\log\left(\frac{c_2}{c_1}\right)\right) \tag{18}$$

## F.1  Proof of Theorem 2

The proof of the almost sure sample complexity result follows naturally from the continuity of $\psi$ (see Lemma 1) and of $C^\star(\mu)$ (see Lemma 2).

We start by defining the event

$$\mathcal{E} = \left\{d_\infty((N_a(t)/t)_{a \in \mathcal{A}}, C^\star(\mu)) \xrightarrow[t \to \infty]{} 0 \text{ and } \hat{\mu}_t \xrightarrow[t \to \infty]{} \mu\right\}.$$

Observe that $\mathcal{E}$ holds with probability 1. This follows from Lemma 3, Lemma 5 and Proposition 1. Let $\varepsilon > 0$. By continuity of $\psi$, there exists an open neighborhood $\mathcal{V}(\varepsilon)$ of $\{\mu\} \times C^\star(\mu)$ such that for all $(\mu', w') \in \mathcal{V}(\varepsilon)$, it holds that

$$\psi(\mu', w') \geq (1 - \varepsilon)\psi(\mu, w^\star),$$

where $w^\star$ is some element in $C^\star(\mu)$. Now, observe that under the event $\mathcal{E}$, there exists $t_0 \geq 1$ such that for all $t \geq t_0$ it holds that $(\hat{\mu}_t, (N_a(t)/t)_{a \in \mathcal{A}}) \in \mathcal{V}(\varepsilon)$, thus for all $t \geq t_0$, it follows that

$$\psi(\hat{\mu}_t, (N_a(t)/t)_{a \in \mathcal{A}}) \geq (1 - \varepsilon)\psi(\mu, w^\star).$$

Since $\hat{\mu}_t \xrightarrow[t \to \infty]{} \mu$ and $a_\mu^\star$ is unique, there exists $t_1 \geq 0$ such that for all $t \geq t_1$, $\hat{a}_t$ is unique. Thus, by Lemma 1, we may write

$$Z(t) = \min_{a \neq a_{\hat{\mu}_t}^*} \frac{\hat{\mu}_t^\top (a_{\hat{\mu}_t}^* - a)^2}{2(a_{\hat{\mu}_t}^* - a)^\top \left(\sum_{s=1}^t a_s a_s^\top\right)^{-1} (a_{\hat{\mu}_t} - a)} = t\psi(\hat{\mu}_t, (N_a(t)/t)_{a \in \mathcal{A}}).$$

By Lemma 5, there exists $t_2 \geq 1$ such that for all $t \geq t_2$ we have

$$\sum_{s=1}^t a_s a_s^\top \succ c I_d.$$

Hence, under the event $\mathcal{E}$, for all $t \geq \max\{t_0, t_1, t_2\}$,

$$Z(t) \geq t(1 - \varepsilon)\psi(\mu, w^\star) \text{ and } \sum_{s=1}^t a_s a_s^\top \succ c I_d.$$

This implies that

$$\tau_\delta = \inf\left\{t \in \mathbb{N}^* : Z(t) > \beta(\delta, t) \quad \text{and} \quad \sum_{s=1}^t a_s a_s^\top \succeq c I_d\right\}$$

$$\leq \max\{t_0, t_1, t_2\} \vee \inf\{t \in \mathbb{N}^* : (1 - \varepsilon)t\psi(\mu, w^\star) > \beta(\delta, t)\}$$

$$\leq \max\{t_0, t_1, t_2\} \vee \inf\left\{t \in \mathbb{N}^* : (1 - \varepsilon)t\psi(\mu, w^\star) > c_1 \log\left(\frac{c_2 t^\gamma}{\delta}\right)\right\}$$

$$\lesssim \max\left\{t_0, t_1, t_2, \frac{1}{1 - \varepsilon} T_\mu^* \log\left(\frac{1}{\delta}\right)\right\},$$

where $c_1, c_2, \gamma$ denote the positive constants independent of $\delta$ and $t$ that appear in the definition of $\beta(t, \delta)$ (see (9)). We used Lemma 8 in the last inequality for $\delta$ sufficiently small. This shows that $\mathbb{P}(\tau_\delta < \infty) = 1$ and in particular that

$$\mathbb{P}\left(\limsup_{\delta \to 0} \frac{\tau_\delta}{\log\left(\frac{1}{\delta}\right)} \lesssim T_\mu^*\right) = 1.$$

$\square$

## F.2 Proof of Theorem 3

Compared to the almost sure result, the expected sample complexity guarantee is more difficult to prove. We break our analysis into three steps. In the first step, we construct a sequence of events over which the stopping time that defines our stopping rule is well-behaved. This requires precise manipulations of the continuity properties of $\psi$ and $C^\star(\mu)$ in combination with the tracking Lemma 6. In the second step, we show indeed that on these events, the stopping time is upper bounded up to a constant by the optimal sample complexity. In the third step, we show that the probabilities of the events under which the sample complexity is not well-behaved are negligible. This is guaranteed thanks to the lazy condition (10) and the sufficient exploration (ensured by Lemma 5 under our sampling rule). We finally conclude by giving the upper bound on the expected sample complexity.

*Proof.* Let $\varepsilon > 0$.

**Step 1.** By continuity of $\psi$ (see Lemma 1), there exists $\xi_1(\varepsilon) > 0$ such that for all $\mu' \in \mathbb{R}^d$ and $w' \in \Lambda$

$$\begin{cases} \|\mu' - \mu\| & \leq \xi_1(\varepsilon) \\ d_\infty(w', C^\star(\mu)) & \leq \xi_1(\varepsilon) \end{cases} \implies |\psi(\mu, w^\star) - \psi(\mu', w')| \leq \varepsilon \psi(\mu, w^\star) = \varepsilon (T_\mu^\star)^{-1} \qquad (19)$$

for any $w^\star \in \arg\min_{w \in C^\star(\mu)} d_\infty(w', w)$ (we have $w^\star \in C^\star(\mu)$). Furthermore, by the continuity properties of the correspondance $C^\star$ (see Lemma 2), there exists $\xi_2(\varepsilon) > 0$ such that for all $\mu' \in \mathbb{R}^d$

$$\|\mu - \mu'\| \leq \xi_2(\varepsilon) \implies \max_{w'' \in C^\star(\mu')} d_\infty(w'', C^\star(\mu)) < \frac{\xi_1(\varepsilon)}{2(K-1)}$$

Let $\xi(\varepsilon) = \min(\xi_1(\varepsilon), \xi_2(\varepsilon))$. In the following, we construct $T_0$, and for each $T \geq T_0$ an event $\mathcal{E}_T$, under which for all $t \geq T$, it holds

$$\|\mu - \hat{\mu}_t\| \leq \xi(\varepsilon) \implies d_\infty((N_a(t)/t)_{a \in \mathcal{A}}, C^\star(\mu)) \leq \xi_1(\varepsilon)$$

Let $T \geq 1$, and define the following events

$$\mathcal{E}_{1,T} = \bigcap_{t=\ell(T)}^{\infty} \{\|\mu - \hat{\mu}_t\| \leq \xi(\varepsilon)\}$$

$$\mathcal{E}_{2,T} = \bigcap_{t=T}^{\infty} \left\{ \inf_{s \geq \ell(t)} d_\infty(w(t), C^\star(\hat{\mu}_s)) \leq \frac{\xi_1(\varepsilon)}{4(K-1)} \right\}$$

$$\subseteq \bigcap_{t=T}^{\infty} \left\{ \exists s \geq \ell(t) : d_\infty(w(t), C^\star(\hat{\mu}_s)) \leq \frac{\xi_1(\varepsilon)}{2(K-1)} \right\}.$$

Note that, under the event $\mathcal{E}_{1,T} \cap \mathcal{E}_{2,T}$, we have for all $t \geq T$, there exists $s \geq \ell(t)$ such that

$$d_\infty(w(t), C^\star(\mu)) \leq d_\infty(w(t), C^\star(\hat{\mu}_s)) + \max_{w' \in C^\star(\hat{\mu}_s)} d_\infty(w', C^\star(\mu))$$

$$< \frac{\xi_1(\varepsilon)}{2(K-1)} + \frac{\xi_1(\varepsilon)}{2(K-1)} = \frac{\xi_1(\varepsilon)}{K-1}$$

Define $\varepsilon_1 = \xi_1(\varepsilon)/(K-1)$. By Lemma 6, there exists $t_1(\varepsilon_1) \geq T$ such that

$$d_\infty((N_a(t)/t)_{a \in \mathcal{A}}, C^\star(\mu)) \leq (p_t + d - 1)\frac{\xi_1(\varepsilon)}{K-1} \leq \xi_1(\varepsilon),$$

and more precisely $t_1(\varepsilon_1) = \max\left\{1/\varepsilon_1^3, 1/(\varepsilon_1^2 d), T/\varepsilon_1^3, 10/\varepsilon_1\right\}$ (see the proof of Lemma 6) where . Thus for $T \geq \max\{10\varepsilon_1^2, \varepsilon_1/d, 1\}$, we have $t_1(\varepsilon_1) = \lceil T/\varepsilon_1^3 \rceil$. Hence, defining for all $T \geq \varepsilon_1^{-3}$, the event

$$\mathcal{E}_T = \mathcal{E}_{1,\lceil \varepsilon_1^3 T \rceil} \cap \mathcal{E}_{2,\lceil \varepsilon_1^3 T \rceil},$$

we have shown that for all $T \geq T_0 = \max(10\varepsilon_1^5, \varepsilon_1^4/d, \varepsilon_1^3, 1/\varepsilon_1^3)$, the following holds

$$\forall t \geq T, \quad \|\mu - \hat{\mu}_t\| \leq \xi(\varepsilon) \implies d_\infty((N_a(t)/t)_{a \in \mathcal{A}}, C^\star(\mu)) \leq \xi_1(\varepsilon). \qquad (20)$$

Finally, combining the implication (20) with the fact that (19) holds under $\mathcal{E}_T$ we conclude that for all $T \geq T_0$, under $\mathcal{E}_T$ we have

$$\psi(\hat{\mu}_t, (N_a(t)/t)_{a \in \mathcal{A}}) \geq (1 - \varepsilon)\psi^\star(\mu). \qquad (21)$$

**Step 2:** Let $T \geq T_0 \vee T_1$ where $T_1$ is defined as

$$T_1 = \inf\left\{t \in \mathbb{N}^* : \lambda_{\min}\left(\sum_{s=1}^{t} a_s a_s^\top\right) \succeq c I_d\right\},$$

where we recall that $c$ is the constant chosen in the stopping rule and is independent of $\delta$. We note that by Lemma 7 for all $t \geq T_1$ we have

$$Z(t) = t\psi(\hat{\mu}_t, (N_a(t)/t)_{a \in \mathcal{A}}).$$

Thus under the event $\mathcal{E}_T$, the inequality (21) holds, and for all $t \geq T$ we have

$$Z(t) > t(1 - \varepsilon)(T_\mu^\star)^{-1}.$$

Under the event $\mathcal{E}_T$, we have

$$\tau = \inf\left\{t \in \mathbb{N}^* : Z(t) > \beta(\delta, t) \text{ and } \sum_{s=1}^{t} a_s a_s^\top \succeq c I_d\right\}$$

$$\leq \inf\{t \geq T : Z(t) > \beta(\delta, t)\}$$

$$\leq T \vee \inf\left\{t \in \mathbb{N}^* : t(1 - \varepsilon)(T_\mu^\star)^{-1} \geq \beta(\delta, t)\right\}$$

$$\leq T \vee \inf\left\{t \in \mathbb{N}^* : t(1 - \varepsilon)(T_\mu^\star)^{-1} \geq c_1 \log(c_2 t^\gamma/\delta)\right\}$$

where $c_1, c_2, \gamma$ are the positive constants that appear in the definition of the threshold $\beta(\delta, t)$ and do not depend on $t$ nor $\delta$ and where we have in particular $c_1 \lesssim \sigma^2$. Applying Lemma 8 yields

$$\inf\left\{t \in \mathbb{N}^* : t(1 - \varepsilon)(T_\mu^\star)^{-1} \geq c_1 \log(c_2 t^\gamma/\delta)\right\} \leq T_2^\star(\delta),$$

where $T_2^\star(\delta) = \frac{c_1}{1-\varepsilon} T_\mu^\star \log(1/\delta) + o(\log(1/\delta))$. This means for $T \geq \max\{T_0, T_1, T_2^\star(\delta)\}$, we have shown that

$$\mathcal{E}_T \subseteq \{\tau \leq T\} \tag{22}$$

Define $T_3^\star(\delta) = \max\{T_0, T_1, T_2^\star(\delta)\}$. We may then write for all $T \geq T_3^\star(\delta)$

$$\tau_\delta \leq \tau_\delta \wedge T_3^\star(\delta) + \tau_\delta \vee T_3^\star(\delta) \leq T_3^\star(\delta) + \tau_\delta \vee T_3^\star(\delta).$$

Taking the expectation of the above inequality, and using the set inclusion (22), we obtain that

$$\mathbb{E}[\tau] \leq T_3^\star(\delta) + \mathbb{E}[\tau \vee T_3^\star(\delta)]$$

Now we observe that

$$\mathbb{E}[\tau \vee T_3^\star(\delta)] = \sum_{T=0}^{\infty} \mathbb{P}(\tau \vee T_3^\star(\delta) > T)$$

$$= \sum_{T=T_3^\star(\delta)+1}^{\infty} \mathbb{P}(\tau \vee T_3^\star(\delta) > T)$$

$$= \sum_{T=T_3^\star(\delta)+1}^{\infty} \mathbb{P}(\tau > T)$$

$$\leq \sum_{T=T_3^\star(\delta)+1}^{\infty} \mathbb{P}(\mathcal{E}_T^c)$$

$$\leq \sum_{T=T_0 \vee T_1}^{\infty} \mathbb{P}(\mathcal{E}_T^c)$$

We have thus shown that

$$\mathbb{E}[\tau] \leq \frac{c_1}{1 - \varepsilon} T_\mu^\star \log(1/\delta) + o(\log(1/\delta)) + T_0 \vee T_1 + \sum_{T=T_0 \vee T_1}^{\infty} \mathbb{P}(\mathcal{E}_T^c). \tag{23}$$

**Step 3:** We now show that $\sum_{T=T_0 \vee T_1 + 1}^{\infty} \mathbb{P}(\mathcal{E}_T^c) < \infty$ and that it can be upper bounded by a constant independent of $\delta$. To ensure this, we shall see that there is a minimal rate by which the sequence $(\ell(t))_{t \geq \infty}$ must grow. Let $T \geq T_0 \vee T_1$, we have by the union bound

$$\mathbb{P}(\mathcal{E}_T^c) \leq \mathbb{P}(\mathcal{E}_{1, \lceil \varepsilon_1^3 T \rceil}^c) + \mathbb{P}(\mathcal{E}_{1, \lceil \varepsilon_1^3 T \rceil}^c).$$

First, using a union bound and the lazy condition (10), we observe that there exists $h\left(\frac{\xi_1(\varepsilon)}{4(K-1)}\right) > 0$ and $\alpha > 0$ such that

$$\mathbb{P}(\mathcal{E}_{1, \lceil \varepsilon_1^3 T \rceil}^c) \leq \sum_{t=\lceil \varepsilon_1^3 T \rceil}^{\infty} \mathbb{P}\left(\inf_{s \geq \ell(t)} d_\infty(w(t), C^\star(\hat{\mu}_s)) > \frac{\xi_1(\varepsilon)}{4(K-1)}\right)$$

$$\leq h\left(\frac{\xi_1(\varepsilon)}{4(K-1)}\right) \sum_{t=\lceil \varepsilon_1^3 T \rceil}^{\infty} \frac{1}{t^{2+\alpha}}$$

$$\leq h\left(\frac{\xi_1(\varepsilon)}{4(K-1)}\right) \int_{\lceil \varepsilon_1^3 T \rceil - 1}^{\infty} \frac{1}{t^{2+\alpha}} dt$$

$$\leq h\left(\frac{\xi_1(\varepsilon)}{4(K-1)}\right) \frac{1}{(1+\alpha)(\lceil \varepsilon_1^3 T \rceil - 1)^{1+\alpha}}.$$

This clearly shows that $\sum_{T=T_0 \vee T_1}^{\infty} \mathbb{P}(\mathcal{E}_{1, \lceil \varepsilon_1^3 T \rceil}^c) < \infty$.

Second, we observe, using a union bound, Lemma 5 and Lemma 4, that there exists strictly positive constants $c_3, c_4$ that are independent of $\varepsilon$ and $T$, and such that

$$\mathbb{P}(\mathcal{E}_{2, \lceil \varepsilon_1^3 T \rceil}^c) \leq \sum_{t=\ell(\lceil \varepsilon_1^3 T \rceil)}^{\infty} \mathbb{P}\left(\|\mu - \hat{\mu}_t\| > \xi(\varepsilon)\right)$$

$$\leq c_3 \sum_{t=\ell(\lceil \varepsilon_1^3 T \rceil)}^{\infty} t^{d/4} \exp(-c_4 \xi(\varepsilon)^2 \sqrt{t}).$$

For $t$ large enough, the function $t \mapsto t^{d/4} \exp(-c_4 \xi(\varepsilon)^2 \sqrt{t})$ becomes decreasing. Additionally, we have by assumption that $(\ell(t))_{t \geq 1}$ is a non decreasing and that $\lim_{t \to \infty} \ell(t) = \infty$, thus we may find $T_2 > T_0 \vee T_1$ such that for all $T \geq T_2$, the function $t \mapsto t^{d/4} \exp(-c_4 \xi(\varepsilon)^2 \sqrt{t})$ is decreasing on $[\ell(\varepsilon_1^3 T) - 1, \infty)$. Hence, for $T \geq T_2$, we have

$$\mathbb{P}(\mathcal{E}_{2, \lceil \varepsilon_1^3 T \rceil}^c) \leq c_3 \int_{\ell(\lceil \varepsilon_1^3 T \rceil) - 1}^{\infty} t^{d/4} \exp(-c_4 \xi(\varepsilon)^2 \sqrt{t}) \, dt.$$

Furthermore, for some $T_3 \geq T_2$ large enough, we may bound the integral for all $T \geq T_3$ as follows

$$\int_{\ell(\lceil \varepsilon_1^3 T \rceil) - 1}^{\infty} t^{d/4} \exp(-c_4 \xi(\varepsilon)^2 \sqrt{t}) \, dt \lesssim \frac{\ell((\lceil \varepsilon_1^3 T \rceil) - 1)^{d/2+1}}{\xi(\varepsilon)^4 \exp\left(c_4 \xi(\varepsilon)^2 \sqrt{\ell(\lceil \varepsilon_1^3 T \rceil) - 1}\right)}.$$

We spare the details of this derivation as the constants are irrelevant in our analysis. Essentially, the integral can be expressed through the upper incomplete Gamma function which can be upper bounded using some classical inequalities [25, 26]. We then obtain that for $T \geq T_3$,

$$\mathbb{P}(\mathcal{E}_{2, \lceil \varepsilon_1^3 T \rceil}^c) \lesssim \frac{\ell((\lceil \varepsilon_1^3 T \rceil) - 1)^{d/2+1}}{\xi(\varepsilon)^4 \exp\left(c_4 \xi(\varepsilon)^2 \sqrt{\ell(\lceil \varepsilon_1^3 T \rceil) - 1}\right)}.$$

Now, the lazy condition (10) ensures that $\lim_{t \to \infty} \ell(t)/t^\gamma > 0$ for some $\gamma \in (0,1)$ and $\ell(t) \leq t$. Thus there exists $T_4 \geq T_3$ such that for all $T \geq T_4$,

$$\mathbb{P}(\mathcal{E}_{2, \lceil \varepsilon_1^3 T \rceil}^c) \lesssim \frac{\ell((\lceil \varepsilon_1^3 T \rceil) - 1)^{d/2+1}}{\xi(\varepsilon)^4 \exp\left(c_4 \xi(\varepsilon)^2 \sqrt{\ell(\lceil \varepsilon_1^3 T \rceil) - 1}\right)} \lesssim \frac{T^{d/2+1}}{\exp\left(c_5(\varepsilon) T^{\gamma/2}\right)}.$$

This shows that

$$
\begin{aligned}
\sum_{T=T_0 \vee T_1}^{\infty} \mathbb{P}(\mathcal{E}_{2,\lceil \varepsilon_1^3 T \rceil}^c) &= \sum_{T=T_0 \vee T_1}^{T_4} \mathbb{P}(\mathcal{E}_{2,\lceil \varepsilon_1^3 T \rceil}^c) + \sum_{T=T_4+1}^{\infty} \mathbb{P}(\mathcal{E}_{2,\lceil \varepsilon_1^3 T \rceil}^c) \\
&\lesssim \sum_{T=T_0 \vee T_1}^{T_4} \mathbb{P}(\mathcal{E}_{2,\lceil \varepsilon_1^3 T \rceil}^c) + \sum_{T=T_4+1}^{\infty} \frac{T^{d/2+1}}{\exp\left(c_5(\varepsilon) T^{\gamma/2}\right)} \\
&< \infty
\end{aligned}
$$

where the last inequality follows from the fact that we can upper bound the infinite sum by a Gamma function, which is convergent as long as $\gamma > 0$.

Finally, we have thus shown that

$$
\sum_{T=T_0 \vee T_1 + 1}^{\infty} \mathbb{P}(\mathcal{E}_T^c) < \infty. \tag{24}
$$

We note that this infinite sum depends on $(\ell(t))_{t \geq 1}$ and $\varepsilon$ only.

**Last step:** Finally, we have shown that for all $\varepsilon > 0$

$$
\mathbb{E}[\tau] \leq \frac{c_1}{1-\varepsilon} T_\mu^\star \log(1/\delta) + o(\log(1/\delta)) + T_0 \vee T_1 + \sum_{T=T_0 \vee T_1}^{\infty} \mathbb{P}(\mathcal{E}_T^c)
$$

where $\sum_{T=T_0 \vee T_1}^{\infty} \mathbb{P}(\mathcal{E}_T^c) < \infty$ and is independent of $\delta$. Hence,

$$
\limsup_{\delta \to 0} \frac{\mathbb{E}[\tau_\delta]}{\log(1/\delta)} \leq \frac{c_1}{1-\varepsilon} T_\mu^\star.
$$

Letting $\varepsilon$ tend to 0 and recalling that $c_1 \lesssim \sigma^2$, we conclude that

$$
\limsup_{\delta \to 0} \frac{\mathbb{E}[\tau_\delta]}{\log(1/\delta)} \lesssim \sigma^2 T_\mu^\star.
$$

$\square$

# G Best-arm identification on the unit sphere

This section is devoted to the proofs of the results related to the best-arm identification problem where the set of arms is the unit sphere $S^{d-1}$. This set is strictly convex so that for any $\mu \in \mathbb{R}^d \backslash \{0\}$, the optimal action $a_\mu^\star$ is unique. We also note that the sphere enjoys the nice following property: for all $\mu \in \mathbb{R}^d$ and for all $a \in S^{d-1}$,

$$\mu^\top (a_\mu^\star - a) = \frac{\|\mu\|}{2} \|a_\mu^\star - a\|^2 \tag{25}$$

We recall that our study is restricted to models with a parameter $\mu$ in $\mathcal{M}(\varepsilon_0)$.

We derive our sample complexity lower bound, presented in Theorem 4, in the next subsection. We then analyze the performance of our stopping rule, and prove Proposition 3. We conclude with the analysis of the sample complexity of our proposed algorithm, and establish Theorem 5.

## G.1 Lower bound – Proof of Theorem 4

As in the case of a finite set of arms, we can derive a lower bound using a change-of-measure argument. The lower bound is obtained as the value of a constrained minimization problem. We get one constraint for each *confusing* parameter. As it turns out, analyzing the resulting constraints is challenging.

The proof consists of 4 steps. In the first step, we write the constraints generated by all confusing parameters. The set of confusing parameters is denoted by $B_\varepsilon(\mu)$. In the second and third steps, we make successive reductions of the set $B_\varepsilon(\mu)$, and hence reduce the number of constraints (yielding looser lower bounds of the sample complexity). At the end of third step, we have restricted our attention to the set of confusing parameters $\mathcal{R}_\varepsilon(\mu)$, and have provided useful properties of these parameters. The last step of the proof exploits these properties to derive the lower bound.

Let $\varepsilon \in (0, \varepsilon_0/5)$, $\delta \in (0,1)$, and $\mu \in \mathcal{M}(\varepsilon_0)$.

**Step 1: Change-of-measure argument.** We start by a direct consequence of the change-of-measure argument (see Lemma 19 [20]). For all $\lambda \in \mathbb{R}^d$,

$$\frac{1}{2\sigma^2} (\mu - \lambda)^\top \mathbb{E} \left[ \sum_{s=1}^\tau a_s a_s^\top \right] (\mu - \lambda) \geq \sup_{\mathcal{E} \in \mathcal{F}_\tau} \mathrm{kl} \left( \mathbb{P}_\mu(\mathcal{E}), \mathbb{P}_\lambda(\mathcal{E}) \right).$$

This result was shown by Soare in [17] and we omit its proof here. Now for all $\mu \in \mathcal{M}(\varepsilon_0)$, define the set $O_\varepsilon(\mu)$ of $\varepsilon$-optimal arms associated with the linear bandit problem parameterized by $\mu$ as

$$O_\varepsilon(\mu) = \left\{ a \in \mathcal{A} : \mu^\top (a_\mu^\star - a) \leq \varepsilon \right\},$$

and the set $B_\varepsilon(\mu)$ of confusing or bad parameters for $\mu$ as

$$B_\varepsilon(\mu) = \left\{ \lambda \in \mathbb{R}^d : O_\varepsilon(\mu) \cap O_\varepsilon(\lambda) = \emptyset \right\}.$$

Note that $B_\varepsilon(\mu)$ is not empty since $\varepsilon < \varepsilon_0$. Now observe that for any $(\varepsilon, \delta)$-PAC algorithm and for all $\lambda \in B_\varepsilon(\mu)$, we have

$$\mathbb{P}_\mu(\hat{a}_\tau \in O_\varepsilon(\mu)^c) \leq \delta \quad \text{and} \quad \mathbb{P}_\lambda(\hat{a}_\tau \in O_\varepsilon(\mu)^c) \geq \mathbb{P}_\lambda(\hat{a}_\tau \in O_\varepsilon(\lambda)) \geq 1 - \delta.$$

Since $\{\hat{a}_\tau \in O_\varepsilon(\mu)^c)\} \in \mathcal{F}_\tau$, by the monotonicity properties of $x \mapsto \mathrm{kl}(x, 1-x)$, we may write, for $\delta \in (0, 1/2]$,

$$\sup_{\mathcal{E} \in \mathcal{F}_\tau} \mathrm{kl} \left( \mathbb{P}_\mu(\mathcal{E}), \mathbb{P}_\lambda(\mathcal{E}) \right) \geq \mathrm{kl}(\delta, 1-\delta).$$

If $\delta \in [1/2, 0)$ we show similarly, using the event $\{\hat{a}_\tau \in O_\varepsilon(\mu)\}$, that

$$\sup_{\mathcal{E} \in \mathcal{F}_\tau} \mathrm{kl} \left( \mathbb{P}_\mu(\mathcal{E}), \mathbb{P}_\lambda(\mathcal{E}) \right) \geq \mathrm{kl}(1-\delta, \delta) = \mathrm{kl}(\delta, 1-\delta).$$

Hence, for any $(\varepsilon, \delta)$-PAC strategy, for all $\lambda \in B_\varepsilon(\mu)$, we have

$$\frac{1}{2} (\mu - \lambda)^\top \mathbb{E} \left[ \sum_{s=1}^\tau a_s a_s^\top \right] (\mu - \lambda) \geq \mathrm{kl}(\delta, 1-\delta). \tag{26}$$

**Step 2: Reductions of $B_\varepsilon(\mu)$.** Finding the most confusing parameters in $B_\varepsilon(\mu)$ is challenging. We restrict our search to a simpler set of confusing parameters at the cost of obtaining a looser bound. First reduction. Define the set

$$\mathcal{D}_\varepsilon(\mu) \triangleq \left\{ \lambda \in \mathcal{M}(\varepsilon_0) : \mu^\top(a_\mu^\star - a_\lambda^\star) > \left(1 + \sqrt{\frac{\|\mu\|}{\|\lambda\|}}\right)^2 \varepsilon \right\}. \tag{27}$$

We prove that $D_\varepsilon(\mu) \subseteq B_\varepsilon(\mu)$. First, let us note that $D_\varepsilon(\mu)$ is non-empty. Indeed, since $\mu \in \mathcal{M}(\varepsilon_0)$, the arm $-a_\mu^\star \notin O_\varepsilon(\mu)$ since $\mu^\top(a_\mu^\star - (-a_\mu^\star)) > 2\varepsilon_0 > 2\varepsilon$. Consider $\lambda = -3\mu = -3\|\mu\|a_\mu^\star$. The optimal arm for $\lambda$ is $-a_\mu^\star$ (because $\mathcal{A} = S^{d-1}$), which gives $(1 + \sqrt{\|\mu\|/\|\lambda\|})^2\varepsilon = (16\varepsilon/9) < 2\varepsilon$. Thus, $\lambda \in \mathcal{D}_\varepsilon(\mu)$.

Now, let $\lambda \in \mathcal{D}_\varepsilon(\mu)$ and let us show that $O_\varepsilon(\mu) \cap O_\varepsilon(\lambda) = \emptyset$. Let $a \in O_\varepsilon(\mu)$, then

$$\langle \lambda, a_\lambda^* - a \rangle = \frac{\|\lambda\|}{2}\|a_\lambda^* - a\|^2 \tag{using (25)}$$

$$\geq \frac{\|\lambda\|}{2}\left| \|a_\lambda^* - a_\mu^*\| - \|a_\mu^* - a\| \right|^2 \tag{reverse triangular inequality}$$

$$= \frac{\|\lambda\|}{\|\mu\|}\left| \sqrt{\frac{\|\mu\|}{2}}\|a_\lambda^* - a_\mu^*\| - \sqrt{\frac{\|\mu\|}{2}}\|a_\mu^* - a\| \right|^2$$

$$= \frac{\|\lambda\|}{\|\mu\|}\left| \sqrt{(\mu, a_\mu^\star - a_\lambda^\star)} - \sqrt{\mu^\top(a_\mu^\star - a)} \right|^2 \tag{using (25)}$$

$$> \frac{\|\lambda\|}{\|\mu\|}\left( \left(1 + \sqrt{\frac{\|\mu\|}{\|\lambda\|}}\right)\sqrt{\varepsilon} - \sqrt{\varepsilon} \right)^2 \tag{since $\lambda \in \mathcal{D}_\varepsilon(\mu)$ and $a \in O_\varepsilon(\mu)$}$$

$$= \varepsilon,$$

thus $a \notin O_\varepsilon(\lambda)$. We have shown that

$$\mathcal{D}_\varepsilon(\mu) \subseteq B_\varepsilon(\mu). \tag{28}$$

Second reduction. Next, we further reduce the set to $\mathcal{H}(\mu) \cap \mathcal{D}_\varepsilon(\mu)$, where $\mathcal{H}(\mu)$ is defined below. Denote by $\mathcal{G}(S^{d-1}, a_\mu^\star)$ the tangent space of $S^{d-1}$ at $a_\mu^\star$. Define

$$\mathcal{H}(\mu) \triangleq \left\{ \lambda \in \mathcal{M}(\varepsilon_0) : \frac{\lambda}{\|\mu\|} \in \mathcal{G}(S^{d-1}, a_\mu^\star) \right\}. \tag{29}$$

Note that if $\lambda \in \mathcal{H}(\mu)$, then $\|\lambda\| \geq \|\mu\|$. This is because on the sphere, it also happens that $a_\mu^\star = \mu/\|\mu\| \in \mathcal{H}(\mu)$ and is the closest point to the origin from $\mathcal{H}(\mu)$. Let us prove that $\mathcal{H}(\mu) \cap \mathcal{D}_\varepsilon(\mu)$ is not empty.

First, let $a \in O_{4\varepsilon}(\mu)$, thus $\varepsilon_0 < \mu^\top a_\mu^\star \leq \mu^\top a + 4\varepsilon$, thus $\mu^\top a > \varepsilon_0 - 4\varepsilon > \varepsilon_0 - 5\varepsilon > 0$, which further implies that $\mu^\top a_\mu^\star - 4\varepsilon > \mu^\top a_\mu^\star - 5\varepsilon > 0$. Hence, by continuity of the map $b \mapsto \mu^\top b$ on the sphere, we may find arms $b \in S^{d-1}$ such that $\mu^\top a_\mu^\star - 4\varepsilon > \mu^\top b > \mu^\top a_\mu^\star - 5\varepsilon > 0$. Thus, for each of these arms, there exists a parameter $\lambda_b \in \mathcal{H}(\mu)$ such that $b = \lambda_b/\|\lambda_b\| = \arg\max_{b \in S^{d-1}} \lambda_b^\top b$. In addition, we have that, for such arms, $5\varepsilon > \mu^\top(a_\mu^\star - b) > 4\varepsilon$, and since $\|\lambda_b\| > \|\mu\|$, we obtain

$$5\varepsilon > \mu^\top(a_\mu^\star - b) > 4\varepsilon > \left(1 + \sqrt{\frac{\|\mu\|}{\|\lambda_b\|}}\right)^2 \varepsilon \tag{30}$$

This shows that $\lambda_b$ belongs to $\mathcal{D}_\varepsilon(\mu)$. Hence $\mathcal{H}(\mu) \cap \mathcal{D}_\varepsilon(\mu)$ is not empty.

**Step 3: Final reduction, and properties.** The final reduction stems from the following observation. From (25), all elements $b \in S^{d-1}$, such that $8\varepsilon/\|\mu\| < \|a_\mu^\star - b\|^2 < 10\varepsilon/\|\mu\|$ have their associated $\lambda_b \in \mathcal{H}(\mu) \cap \mathcal{D}_\varepsilon(\mu)$. We denote by $\mathcal{R}_\varepsilon(\mu)$ the corresponding set of parameters:

$$\mathcal{R}_\varepsilon(\mu) \triangleq \{\lambda \in \mathcal{H}(\mu) \cap \mathcal{D}_\varepsilon(\mu) : 4\varepsilon < \mu^\top(a_\mu^\star - a_\lambda^\star) < 5\varepsilon\}. \tag{31}$$

Note that the span of the set $\{\lambda - \mu : \ \lambda \in \mathcal{R}_\varepsilon(\mu)\}$ is a $d-1$-dimensional space.

Next, we establish the following useful property. There are constants $c_1, c_2 > 0$ such that for any $\lambda \in \mathcal{R}_\varepsilon(\mu)$,

$$c_1 \|\mu\| \varepsilon \leq \|\lambda - \mu\|^2 \leq c_2 \|\mu\| \varepsilon.$$

To this aim, we first establish, using elementary geometry, the following identity for all $\lambda \in \mathcal{H}(\mu)$

$$\|\mu - \lambda\|^2 (\|\mu\| - \Delta(a_\lambda^\star))^2 + \|\mu\|^2 \Delta(a_\lambda^\star)^2 = \|\mu\|^4 \|a_\mu^\star - a_\lambda^\star\|^2 \tag{32}$$

where $\Delta(a) = \mu^\top(a_\mu^\star - a)$ denotes the gap between $a$ and the best arm. To show the identity (32), let us note that $\mu, \lambda$ and $0$ (the center of the sphere $S^{d-1}$) define a 2-dimensional plane, and that $a_\mu^\star$ and $a_\lambda^\star$ belong to this plane. Without loss of generality, we may assume that $\|\mu\| = 1$ (we can always renormalize). Since $\mu, \lambda \in \mathcal{H}(\mu)$, and by construction $(\mu/\|\mu\|)^\top (\mu - \lambda) = a_\mu^{\star\top}(\mu - \lambda) = 0$. Thales' Theorem (the intercept Theorem) guarantees

$$\frac{\Delta(a_\mu^\star)}{1} = \frac{\|p - \lambda\|}{\|\mu - \lambda\|},$$

where $p$ is the orthogonal projection of $a_\lambda^\star$ on $\mathcal{H}(\mu)$. Next, by Pythagoras' Theorem, we have

$$\|\mu - p\|^2 + \Delta(a_\lambda^\star)^2 = \|a_\mu^\star - a_\lambda^\star\|^2.$$

By construction, we have $\|\mu - \lambda\| = \|\mu - p\| + \|p - \lambda\|$, and using the above two equations gives

$$\|\mu - \lambda\|^2 (1 - \Delta_{\mu,\mathcal{A}}(a_\lambda^*))^2 + \Delta_{\mu,\mathcal{A}}(a_\lambda^*)^2 = \|a_\mu^* - a_\lambda^*\|^2,$$

which gives (32) by just renormalizing. Now, If follows immediately from (25) and (32) that

$$\|\mu - \lambda\|^2 = \|\mu\|^2 \|a_\mu^\star - a_\lambda^\star\|^2 \frac{4 - \|a_\mu^\star - a_\lambda^\star\|^2}{(2 - \|a_\mu^\star - a_\lambda^\star\|^2)^2}.$$

Note that on the sphere for $\lambda \in \mathcal{H}(\mu)$, we have $0 \leq \|a_\mu^\star - a_\lambda^\star\| \leq 1$. Hence, we obtain

$$\frac{3}{2} \|\mu\|^2 \|a_\mu^\star - a_\lambda^\star\|^2 \leq \|\mu - \lambda\|^2 \leq 4 \|\mu\|^2 \|a_\mu^\star - a_\lambda^\star\|^2,$$

or equivalently, using (25), that

$$3 \|\mu\| \langle \mu, a_\mu^\star - a_\lambda^\star \rangle \leq \|\mu - \lambda\|^2 \leq 8 \|\mu\| \langle \mu, a_\mu^\star - a_\lambda^\star \rangle.$$

Finally let $\lambda \in \mathcal{R}_\varepsilon(\mu) \subseteq \mathcal{H}(\mu) \cap \mathcal{D}_\varepsilon(\mu)$. Since (31) holds, it follows that for such $\lambda$, we have

$$12 \|\mu\| \varepsilon \leq \|\lambda - \mu\|^2 \leq 40 \|\mu\| \varepsilon. \tag{33}$$

**Step 4:** For $\lambda \in \mathcal{R}_\varepsilon(\mu)$, combining satisfying (33) and (26), we obtain

$$\begin{aligned}
\mathrm{kl}(\delta, 1 - \delta) &\leq \frac{1}{2\sigma^2} \inf_{\lambda \in B_\varepsilon(\mu)} (\mu - \lambda)^\top \mathbb{E}\left[\sum_{s=1}^{\tau} a_s a_s^\top\right] (\mu - \lambda) \\
&\leq \frac{1}{2\sigma^2} \inf_{\lambda \in R_\varepsilon(\mu)} (\mu - \lambda)^\top \mathbb{E}\left[\sum_{s=1}^{\tau} a_s a_s^\top\right] (\mu - \lambda) \\
&\leq \frac{1}{2\sigma^2} \inf_{x \in \bar{S}(\mu)} x^\top \mathbb{E}\left[\sum_{s=1}^{\tau} a_s a_s^\top\right] x \|\lambda - \mu\|^2 \\
&\leq \frac{20 \|\mu\| \varepsilon}{\sigma^2} \inf_{x \in \bar{S}(\mu)} x^\top \mathbb{E}\left[\sum_{s=1}^{\tau} a_s a_s^\top\right] x,
\end{aligned}$$

where

$$\bar{S}(\mu) \triangleq \left\{ \frac{\lambda - \mu}{\|\lambda - \mu\|} : \ \lambda \in \mathcal{R}_\varepsilon(T) \right\}$$

Hence, we have shown that

$$\inf_{x \in \bar{S}(\mu)} x^\top \mathbb{E}\left[\sum_{s=1}^{\tau} a_s a_s^\top\right] x \geq \frac{\sigma^2}{20\|\mu\|\varepsilon} \mathrm{kl}(\delta, 1 - \delta). \tag{34}$$

To complete the derivation, we analyze the right hand side of the lower bound (34). First, define the set of sampling rules as follows

$$\mathcal{X} \triangleq \{(a_t)_{t\geq 1} : \forall t \geq 1, \ a_t \text{ is } \mathcal{F}_{t-1}\text{-measurable}\}, \tag{35}$$

and the expected matrix of exploration under a sampling rule $(a_t)_{t\geq 1} \in \mathcal{X}$ as

$$G_\tau((a_t)_{t\geq 1}) \triangleq \mathbb{E}\left[\sum_{s=1}^{\tau} a_s a_s^\top\right].$$

We will show that

$$\sup_{(a_t)_{t\geq 1} \in \mathcal{X}} \inf_{x \in \bar{S}(\mu)} x^\top G_\tau((a_t)_{t\geq 1}) x \leq \frac{\mathbb{E}[\tau]}{d - 1}. \tag{36}$$

For a given symmetric matrix $A \in \mathbb{R}^{d \times d}$, we denote the eigenvalues of $A$ in decreasing order as $\lambda_1(A), \lambda_2(A), \dots, \lambda_d(A)$.

Let $(a_t)_{t\geq 1} \in \mathcal{X}$. We start by noting that $G_\tau((a_t)_{t\geq 1})$ is positive semi-definite matrix and that $\dim(\mathrm{span}(\bar{S})) = d - 1$, therefore, using the Courant-Fisher min-max theorem, we have

$$\lambda_{d-1}(G_\tau((a_t)_{t\geq 1})) \geq \inf_{x \in \bar{S}(\mu)} x^\top G_\tau((a_t)_{t\geq 1}) x \geq 0.$$

Additionally, we observe that for all $t \geq 1$, $\|a_t\| = 1$ since $a_t$ is taking values in $S^{d-1}$. Thus, we obtain

$$\sum_{k=1}^{d} \lambda_k(G_\tau((a_t)_{t\geq 1})) = \mathrm{tr}(G_\tau((a_t)_{t\geq 1})) = \mathbb{E}\left[\sum_{s=1}^{\tau} \|a_s\|^2\right] = \mathbb{E}[\tau],$$

where we used the linearity of the trace and of the expectation. We conclude from the above that the value of max-min optimization problem $\sup_{(a_t)_{t\geq 1} \in \mathcal{X}} \inf_{x \in \bar{S}(\mu)} x^\top G_\tau((a_t)_{t\geq 1}) x$ can be upper bounded by the value of the following optimization problem

$$\max_{\lambda_1, \dots, \lambda_d} \quad \lambda_{d-1}$$

$$\text{s. t.} \quad \sum_{k=1}^{d} \lambda_k = \mathbb{E}[\tau]$$

$$\lambda_1 \geq \lambda_2 \geq \dots \geq \lambda_d \geq 0.$$

We easily see that the value of this optimization problem is $\mathbb{E}[\tau]/(d-1)$ (with $\lambda_d = 0$ and $\lambda_i = \mathbb{E}[\tau]/(d-1)$ for all $i \neq d$). Hence (36) holds.

From (34) and (36), we conclude that

$$\mathbb{E}[\tau] \geq \frac{\sigma^2(d-1)}{40\|\mu\|\varepsilon} \mathrm{kl}(\delta, 1 - \delta).$$

$\square$

## G.2  Stopping rule – Proof of Proposition 3

Let us consider the events

$$\mathcal{E}_1 = \{\tau < \infty\} = \left\{\exists t \in \mathbb{N}^* : Z(t) \geq \beta(\delta, t) \text{ and } \lambda_{\min}\left(\sum_{s=1}^{t} a_s a_s^\top\right) \geq \max\left\{c, \frac{\rho(\delta, t)}{\|\hat{\mu}_t\|^2}\right\}\right\},$$

$$\mathcal{E}_2 = \{\mu^\top(a_\mu^\star - \hat{a}_\tau) > \varepsilon\},$$

$$\mathcal{E}_3 = \bigcap_{t=1}^{\infty} \left\{\|\hat{\mu}_t - \mu\|^2 \leq \left(\frac{\varepsilon}{\varepsilon_t} - 1\right)^2 \frac{\rho(t, \delta_t)}{\lambda_{\min}\left(\sum_{s=1}^{t} a_s a_s^\top\right)} \text{ or } \lambda_{\min}\left(\sum_{s=1}^{t} a_s a_s^\top\right) \geq c\right\}.$$

If there exists $t \geq 1$ such that $\sum_{s=1}^{t} a_s a_s^\top \succ 0$, we have by Lemma 7 that $Z(t) = \inf_{\{b \in \mathcal{A}: |\hat{\mu}_t^\top(\hat{a}_t - b)| \geq \varepsilon_t\}} Z_{\hat{a}_t, b, \varepsilon_t}(t)$ where

$$Z_{\hat{a}_t, b, \varepsilon_t}(t) = \mathrm{sgn}(\hat{\mu}_t^\top(\hat{a}_t - b) + \varepsilon_t) \frac{(\hat{\mu}_t^\top(\hat{a}_t - b) + \varepsilon_t)^2}{2(\hat{a}_t - b)^\top \left(\sum_{s=1}^{t} a_s a_s^\top\right)^{-1}(\hat{a}_t - b)}.$$

Thus, we have

$$\mathcal{E}_1 \cap \mathcal{E}_2 = \left\{ \exists t \in \mathbb{N}^* : \inf_{\{b \in \mathcal{A}: |\hat{\mu}_t^\top(\hat{a}_t - b)| \geq \varepsilon_t\}} Z_{a, b, \varepsilon_t}(t) \geq \beta(\delta, t) \right.$$

$$\left. \text{and } \lambda_{\min}\left(\sum_{s=1}^{t} a_s a_s^\top\right) \geq \max\left\{c, \frac{\rho(\delta, t)}{\|\hat{\mu}_t\|^2}\right\} \text{ and } \mu^\top(a_\mu^\star - \hat{a}_\tau) > \varepsilon \right\}$$

Now using (25), we have $\mu^\top(a_\mu^\star - b) = \frac{\|\mu\|}{2}\|a_\mu^\star - a\|^2$. Thus

$$\mu^\top(a_\mu^\star - \hat{a}_t) > \varepsilon \implies \hat{\mu}_t^\top(\hat{a}_t - a_\mu^\star) = \frac{\|\hat{\mu}_t\|}{\|\mu\|}\mu^\top(a_\mu^\star - \hat{a}_t) > \frac{\|\hat{\mu}_t\|}{\|\mu\|}\varepsilon.$$

Observe that

$$\begin{cases} \|\hat{\mu}_t - \mu\|^2 \leq \left(\frac{\varepsilon}{\varepsilon_t} - 1\right)^2 \frac{\rho(t, \delta_t)}{\lambda_{\min}\left(\sum_{s=1}^{t} a_s a_s^\top\right)} \\ \|\hat{\mu}_t\|^2 \geq \frac{\rho(t, \delta_t)}{\lambda_{\min}\left(\sum_{s=1}^{t} a_s a_s^\top\right)} \end{cases} \implies \left(\frac{\varepsilon}{\varepsilon_t} - 1\right)\|\hat{\mu}_t\| \geq \|\hat{\mu}_t - \mu\|$$

$$\implies \left(\frac{\varepsilon}{\varepsilon_t} - 1\right)\|\hat{\mu}_t\| \geq |\|\hat{\mu}_t\| - \|\mu\||$$

$$\implies \frac{\varepsilon}{\varepsilon_t}\|\hat{\mu}_t\| \geq \|\mu\|.$$

Hence, we have

$$\begin{cases} \mu^\top(a_\mu^\star - \hat{a}_t) > \varepsilon \\ \|\hat{\mu}_t - \mu\|^2 \leq \left(\frac{\varepsilon}{\varepsilon_t} - 1\right)^2 \frac{\rho(t, \delta_t)}{\lambda_{\min}\left(\sum_{s=1}^{t} a_s a_s^\top\right)} \\ \|\hat{\mu}_t\|^2 \geq \frac{\rho(t, \delta_t)}{\lambda_{\min}\left(\sum_{s=1}^{t} a_s a_s^\top\right)} \end{cases} \implies \hat{\mu}_t^\top(\hat{a}_t - a_\mu^\star) > \varepsilon_t$$

It then follows that

$$\mathcal{E}_1 \cap \mathcal{E}_2 \cap \mathcal{E}_3 \subseteq \left\{ t \in \mathbb{N}^* : Z_{\hat{a}_t, a_\mu^\star, \varepsilon_t} \geq \beta(\delta, t) \text{ and } \lambda_{\min}\left(\sum_{s=1}^{t} a_s a_s^\top\right) \geq c \text{ and } \hat{\mu}_t^\top(\hat{a}_t - a_\mu^\star) > \varepsilon_t \right\}.$$

Considering (15), we have under the event $\mathcal{E}_1 \cap \mathcal{E}_2 \cap \mathcal{E}_3$ that

$$\max_{\{\mu':(\mu')^\top(\hat{a}_t - a_\mu^\star) + \varepsilon_t \geq 0\}} f_{\mu'}(r_t, a_t, \ldots, r_1, a_1) = f_{\hat{\mu}_t}(r_t, a_t, \ldots, r_1, a_1),$$

$$\max_{\{\mu':(\mu')^\top(\hat{a}_t - a_\mu^\star) + \varepsilon_t \leq 0\}} f_{\mu'}(r_t, a_t, \ldots, r_1, a_1) \geq f_\mu(r_t, a_t, \ldots, r_1, a_1).$$

As a consequence, under $\mathcal{E}_1 \cap \mathcal{E}_2 \cap \mathcal{E}_3$, we have

$$Z_{\hat{a}_t, a_\mu^\star, \varepsilon_t}(t) = \log\left(\frac{\max_{\mu':(\mu')^\top(\hat{a}_t - a_\mu^\star) + \varepsilon_t \geq 0} f_{\mu'}(r_t, a_t, \ldots, r_1, a_1)}{\max_{\mu':(\mu')^\top(\hat{a}_t - a_\mu^\star) + \varepsilon_t \leq 0} f_{\mu'}(r_t, a_t, \ldots, r_1, a_1)}\right)$$

$$\leq \log\left(\frac{f_{\hat{\mu}_t}(r_t, a_t, \ldots, r_1, a_1)}{f_\mu(r_t, a_t, \ldots, r_1, a_1)}\right)$$

$$= \frac{1}{2}(\hat{\mu}_t - \mu)^\top \left(\sum_{s=1}^{t} a_s a_s^\top\right)(\hat{\mu}_t - \mu)$$

$$= \frac{1}{2}\|\mu - \hat{\mu}_t\|^2_{\sum_{s=1}^{t} a_s a_s^\top},$$

Hence,

$$\mathcal{E}_1 \cap \mathcal{E}_2 \cap \mathcal{E}_3 \subseteq \left\{ \exists t \in \mathbb{N}^* : \frac{1}{2} \|\mu - \hat{\mu}_t\|^2_{\sum_{s=1}^t a_s a_s^\top} \geq \beta(\delta, t) \text{ and } \sum_{s=1}^t a_s a_s^\top \right\}.$$

We further deduce that

$$\mathbb{P}\left(\mathcal{E}_1 \cap \mathcal{E}_2 \cap \mathcal{E}_3\right) \leq \mathbb{P}\left( \exists t \in \mathbb{N}^* : \frac{1}{2} \|\sum_{s=1}^t a_s \eta_s\|^2_{(\sum_{s=1}^t a_s a_s^\top + cI_d)^{-1}} \geq 2\sigma^2 \zeta_t \right)$$

$$\leq \sum_{t=1}^n \mathbb{P}\left( \frac{1}{2} \|\sum_{s=1}^t a_s \eta_s\|^2_{(\sum_{s=1}^t a_s a_s^\top + cI_d)^{-1}} \geq 2\sigma^2 \zeta_t \right)$$

$$\leq \sum_{t=1}^\infty \frac{\delta_t}{2} \leq \frac{\delta}{2},$$

where for the third inequality, we use the result of Proposition 4. Using a union bound and Proposition 4 again, we also have

$$\mathbb{P}\left(\mathcal{E}_3^c\right) \leq \sum_{t=1}^\infty \mathbb{P}\left( \|\mu_t - \mu\|^2 \geq \left(\frac{\varepsilon}{\varepsilon_t} - 1\right)^2 \frac{\rho(t, \delta_t)}{\lambda_{\min}\left(\sum_{s=1}^t a_s a_s^\top\right)}, \sum_{s=1}^t a_s a_s^\top \succeq c \right)$$

$$\leq \sum_{t=1}^\infty \mathbb{P}\left( \|\sum_{s=1}^t a_s \eta_s\|^2_{(\sum_{s=1}^t a_s a_s^\top + cI_d)^{-1}} \geq 2\sigma^2 \zeta_t \right)$$

$$\leq \sum_{t=1}^\infty \frac{\delta_t}{2} \leq \frac{\delta}{2}$$

Finally, we obtain

$$\mathbb{P}\left(\tau < \infty, \mu^\top(a_\mu^\star - \hat{a}_\tau) > \varepsilon\right) = \mathbb{P}(\mathcal{E}_1 \cap \mathcal{E}_2) \leq \mathbb{P}(\mathcal{E}_1 \cap \mathcal{E}_2 \cap \mathcal{E}_3) + \mathbb{P}(\mathcal{E}_3^c) \leq \delta. \qquad (37)$$

$\square$

### G.3 Sample complexity – Proof of Theorem 5

We recall that $\mathcal{U} = \{u_1, \ldots, u_d\}$ is an orthonormal basis in $\mathbb{R}^d$, $\mathcal{U} \subset S^{d-1}$ and our sampling rule is

$$a_t = u_{(t \mod d)}$$

**Almost sure guarantees.** Observe that for all $t \geq d$

$$\left\lceil \frac{t}{d} \right\rceil \sum_{u \in \mathcal{U}} uu^\top \succeq \sum_{s=1}^t a_s a_s^\top \succeq \left\lfloor \frac{t}{d} \right\rfloor \sum_{u \in \mathcal{U}} uu^\top \succ 0. \qquad (38)$$

Let $t \geq d$. We have

$$Z(t) = \inf_{\{b \in \mathcal{A}: |\hat{\mu}_t^\top(\hat{a}_t - b)| \geq \varepsilon_t\}} Z_{\hat{a}_t, b, \varepsilon_t}(t)$$

$$\geq \inf_{\{b \in \mathcal{A}: |\hat{\mu}^\top(\hat{a}_t - b)| \geq \varepsilon_t\}} \frac{(\hat{\mu}_t^\top(\hat{a}_t - b) + \varepsilon_t)^2}{2\|\hat{a}_t - b\|^2} \lambda_{\min}\left(\sum_{s=1}^t a_s a_s^\top\right)$$

$$\geq \inf_{\{b \in \mathcal{A}: |\hat{\mu}_t^\top(\hat{a}_t - b)| \geq \varepsilon_t\}} \left( \frac{\mu_t^\top(\hat{a}_t - b)}{\|\hat{a}_t - b\|} + \frac{\varepsilon_t}{\|\hat{a}_t - b\|} \right)^2 \lambda_{\min}\left(\sum_{s=1}^t a_s a_s^\top\right)$$

$$\geq \inf_{\{b \in \mathcal{A}: |\hat{\mu}_t^\top(\hat{a}_t - b)| \geq \varepsilon_t\}} \left( \frac{\|\mu_t\|}{2} \|\hat{a}_t - b\| + \frac{\varepsilon_t}{\|\hat{a}_t - b\|} \right)^2 \lambda_{\min}\left(\sum_{s=1}^t a_s a_s^\top\right)$$

$$\geq \inf_{\{b \in \mathcal{A}: \|\|\hat{\mu}_t\|\|\hat{a}_t - b)\|^2 \geq 2\varepsilon_t\}} \left( \frac{\|\mu_t\|}{2} \|\hat{a}_t - b\| + \frac{\varepsilon_t}{\|\hat{a}_t - b\|} \right)^2 \lambda_{\min}\left(\sum_{s=1}^t a_s a_s^\top\right)$$

$$\geq 2\varepsilon_t \|\hat{\mu}_t\| \lambda_{\min}\left(\sum_{s=1}^t a_s a_s^\top\right)$$

Thus, using (38), we obtain

$$Z(t) \geq 2\varepsilon_t \|\hat{\mu}_t\| \left\lfloor \frac{t}{d} \right\rfloor. \tag{39}$$

Now, consider the choice

$$\varepsilon_t = \frac{\varepsilon}{1 + \varepsilon \left( 4\sigma^2 \log \left( \frac{4}{\delta_t} \left\lceil \frac{t}{d} \right\rceil \right) \right)^{-1/2}}. \tag{40}$$

Note that for all $\varepsilon_t < \varepsilon$ and $\varepsilon_t \xrightarrow[t \to \infty]{} \varepsilon$. We have

$$\tau \leq d \vee \inf \left\{ t \in \mathbb{N}^* : \quad \varepsilon \|\mu_t\| \left\lfloor \frac{t}{d} \right\rfloor \geq 4\sigma^2 \log \left( \frac{4}{\delta_t} \left\lceil \frac{t}{d} \right\rceil \right) \right\}$$

Now by the force exploration (38), and using (3), we have that $\|\hat{\mu}_t\| \xrightarrow[t \to \infty]{} \|\mu\|$ (a.s.). Define the event $\mathcal{E} = \{\hat{\mu}_t \| \xrightarrow[t \to \infty]{} \|\mu\|\}$. On this event, for all $\xi > 0$, there exists $t_0 > 0$ such that $\|\hat{\mu}_t\| > (1 - \xi)\|\mu\|$. Hence on $\mathcal{E}$, we have

$$\tau \leq \max\{d, t_0\} \vee \inf \left\{ t \in \mathbb{N}^* : \quad \varepsilon(1 - \xi)\|\mu\| \left\lfloor \frac{t}{d} \right\rfloor \geq 4\sigma^2 \log \left( \frac{4}{\delta_t} \left\lceil \frac{t}{d} \right\rceil \right) \right\}.$$

Using Lemma 8 and similar arguments as in the analysis of the sample complexity for the case of finite sets of arms in Appendix F, we obtain that on $\mathcal{E}$,

$$\tau \lesssim \max\{d, t_0\} + \frac{4\sigma^2 d}{(1 - \xi)\|\mu\|} \log \left( \frac{1}{\delta} \right) + o \left( \log \left( \frac{1}{\delta} \right) \right)$$

Thus, we have shown that $\mathbb{P}(\tau < \infty) = 1$ and more precisely, letting $\xi$ tend to 0, that

$$\mathbb{P} \left( \limsup_{\delta \to 0} \frac{\tau}{\log(1/\delta)} \lesssim \frac{\sigma^2 d}{\varepsilon \|\mu\|} \right) = 1 \tag{41}$$

**Guarantees in expectation.** To obtain an upper bound on the expected sample complexity, we construct for all $T \geq 1$, the events

$$\mathcal{E}_T = \bigcap_{t=T}^{\infty} \{\|\hat{\mu}_t - \mu\| \leq \xi\|\mu\|\} \tag{42}$$

Following the same chain of arguments as in Appendix F.2 (see Step 2), we can show that

$$\mathbb{E}[\tau] \lesssim \frac{d\sigma^2}{(1 - \xi)\varepsilon\|\mu\|} \log(1/\delta) + o(\log(1/\delta)) + d + \sum_{T=d}^{\infty} \mathbb{P}(\mathcal{E}_T^c). \tag{43}$$

Then again using the forced exploration (38) and Lemma 4, we obtain that for all $T \geq 1$

$$\mathbb{P}(\mathcal{E}_T^c) \leq \sum_{t=T}^{\infty} c_1 \exp(-c_2 \xi^2 \|t), \tag{44}$$

where $c_1, c_2$ are positive constants that only depends on $d, \mu$ and $\sigma$. Then following similar steps as in Appendix F.2 (see Step 3), we can show that $\sum_{T=d}^{\infty} \mathbb{P}(\mathcal{E}_T^c) < \infty$, from which we may then conclude that

$$\limsup_{\delta \to 0} \frac{\mathbb{E}[\tau]}{\log(1/\delta)} \lesssim \frac{d\sigma^2}{\|\mu\|\varepsilon}.$$

$\square$

## Footnotes

[2]We mean by $A^{-1}$ the pseudo-inverse of $A$ when the matrix is not invertible.