[Reviews · NeurIPS 2020]

Review 1

Summary and Contributions: This paper studies best arm identification in linear bandits in the fixed confidence setting. The authors provide an asymptotically optimal algorithm and evaluate it through experiments. They also analyze a best arm identification problem on the unit sphere.

Strengths: The proposed algorithm is simple and the analysis in the paper is rigorous. It does seem relevant to the pure exploration bandits community to have a track-and-stop algorithm for linear bandits.

Weaknesses: The authors claim that their experiments suggest that their algorithm has a strong empirical performance. I am unconvinced because they only run experiments on one set up. In order to support this claim, they need to consider a variety of experimental setups. One of the central claims in this paper is they provide a practical, asymptotically optimal algorithm, so it is problematic that they do not provide more support demonstrating its practicability. It does not seem very difficult to come up with a practical, yet asymptotically optimal algorithm (up to constants). One can combine the RAGE algorithm with the explore-and-then-verify framework of [13] (see "Nearly Optimal Sampling Algorithms for Combinatorial Pure Exploration" by Chen et al. for an example of this sort of algorithm in the setting of combinatorial bandits). It seems like the contribution here is to get rid of a constant and to come up with an algorithm in the style of [18]. Regarding their analysis of \epsilon-good arm identification for the sphere, it seems that one could just construct an (\epsilon/2)-net of the sphere and apply an algorithm for finite number of bandits to this. Is there anything fundamentally more difficult about this problem setting? It is unclear how often to update the allocation in practice. If this confers computational advantages in practice, it is important to provide practitioners with guidance on this.

Correctness: Yes.

Clarity: The paper is clear.

Relation to Prior Work: Yes.

Reproducibility: Yes

Additional Feedback: It would be useful if the authors clarified what made solving this problem challenging.


Review 2

Summary and Contributions: The paper considers the best arm identification problem in linear bandits. It proposes the track-and-stop type algorithm and proves it achieves theoretical lower bound. The empirical results seem promising as well.

Strengths: It proposes a novel algorithm for BAI in linear bandits, which is an important problem. The result of achieving theoretical lower bound is significant. As far as I checked, the proof seems correct.

Weaknesses: To me, there is no explicit weakness.

Correctness: I could not find any mistake in the proof

Clarity: Yes

Relation to Prior Work: Yes

Reproducibility: Yes

Additional Feedback:


Review 3

Summary and Contributions: This work gives a quite clean and asymptotically optimal answer for best arm identification in linear bandits. It builds upon the track-and-stop framework of Garivier and Kaufmann, with the interesting twist that the authors consider here a lazy version (which does not hurt the asymptotic optimality).

Strengths: Best arm identification in linear bandits has potentially many applications, and the proposed algorithm has both a solid theoretical guarantee and seems to perform well in practice.

Weaknesses: The problem is not exactly ``fresh", with already many papers on the same exact topic. The same can be said about the proposed technique, which by now is getting quite classical. In some ways it is a bit frustrating to see an asymptotic analysis, when arguably the field went through a resurgence when it was realized that finite-time guarantees actually make a difference (I am referring to Auer et al. versus Lai and Robbins type analyses).

Correctness: The paper appears to be correct.

Clarity: The paper is well written, and I especially appreciated the very lucid presentation in Section 3.

Relation to Prior Work: The prior work is appropriate.

Reproducibility: Yes

Additional Feedback:


Review 4

Summary and Contributions: I thank the author for the rebuttal. I still think this paper could be valuable for the community. However after discussing with the other reviewers and the meta reviewer we would have liked a discussion on the validity (or not) of the Franke-Wolfe algorithm to update the allocation (see the recent paper by Degenne for instance) and a better discussion and more details on the complexity not depending on the number of arms. This paper introduces a new strategy for the problem of best arm identification with fixed confidence in stochastic linear bandits. This strategy allows to achieve the known lower bound asymptotically almost surely and in expectation (main results of the paper). The strategy is based on the track and stop principle: it tracks the optimal distribution over the arms using the estimated regression parameter. This optimal proportion can be updated as often as one wants. The authors also provide a first result for best arm identification in linear bandits for a continuous set of arms compared to the usual finite set of arms. Finally n experiment compares the introduced algorithm Lazy TS with the state-of-the-art RAGE algorithm.

Strengths: The approach devised by the authors is novel compared to existing work. Most existing work relies on rounding procedures which implies additional computational costs and does not reach the lower bound when the confidence level delta tends to 0 (e.g. XY-adaptive allocation from Soare et al, RAGE from Fiez et al. and Tao et al. (does not reach lower bound but does not use a rounding procedure)). A very recent paper, "Gamification of Pure Exploration for Linear Bandits" by Degenne et al., presented at ICML 2020. https://proceedings.icml.cc/static/paper_files/icml/2020/4512-Supplemental.pdf also obtains an asymptotically optimal algorithm for fixed-confidence pure exploration in linear bandits but the strategy derived therein is different from the one of this paper. The authors might want to cite this paper. The introduced algorithm, Lazy TS, seems to perform well in practice on the experiment considered in the paper

Weaknesses: More classical experiments usually considered for best arm identification in linear bandits could be done in order to better assess the performance of the algorithm in practice compared to existing algorithms (see eg the experiments in the RAGE paper). How bad can be the c_{A_0} constant?

Correctness: The claims look correct but I did not check the proofs in the supplementary material.

Clarity: Overall the paper is well written. I had sometimes a hard time connecting the different results of section 3.3. which describe how the sampling rule is designed. Some terms are not very clear to me such as "certainty equivalence principle".

Relation to Prior Work: The related work and the advantages of the proposed strategy is clearly discussed. As mentioned above only one very recent paper deriving the same result but with a different strategy is missing.

Reproducibility: Yes

Additional Feedback: Could the authors describe the idea of the track and stop principle when mentioning it in the introduction as this is one of the most important tools of the strategy derived in the paper? This would make the paper better self contained. Usually in best arm identification we care about the performance of the estimator hat mu in the directions (a - a') and we use an upper bound of <hat mu - mu, a -a'> as we do not need to be good in all the directions to discriminate arms and find the best one. These directions appear in the stopping rule and in the lower bound but not in the bound use for the least square estimator. Can the authors detail why they do not need such a bound compared to existing work? Can you confirm that the goal of A_0 is to prevent the minimum eigenvalue of the design matrix to be too small (which could be arbitrarily small depending on the set of arms and the optimal distributions)? However it seems to me that c_{A_0} could still be very small in practice: for instance in R^2, if the set of arms is made of 2 arms with same norm and the angle between these 2 arms tends to 0. I would like an interpretation of the stopping rule. For the continuous set of arms why does one need to restrict the study to the unit sphere? Why does the continuous set make the analysis challenging as written l 243? - l 130: do we really need it for the paper without the supplementary materials? - l 141: L appears twice: L = max ... <= L - a is both used to denote a member of the set of arms and an integer in {1, ..., K}. - A_0 is not specified in Algorithm 1. In the experiments it is chosen at random but is it really the best strategy? Shouldn't one pick the best A_0 if possible? - I would use V instead of A in Algorithm 1 to denote the design matrix. - The algorithm called Lazy Track and Stop is referred to as LTS at some places and as Lazy TS in other places. - l 249: be a subset, that forms - l 176 : Franke Broader impact: please also state that recommender systems can be used with bad intention such as influence people opinion during elections. A lot of people have been raising issues of such systems lately.

[Author Response · NeurIPS 2020]

We thank the reviewers for their detailed and constructive comments. Please find our answers below.

| Algorithm | Lazy TS | RAGE | Oracle | | Algorithm | Lazy TS | RAGE | $\mathcal{XY}$-Adaptive | Oracle |
|---|---|---|---|---|---|---|---|---|---|
| dimension | Mean (Std) | Mean (Std) | Mean (Std) | | confidence level | Mean (Std) | Mean (Std) | Mean (Std) | Mean (Std) |
| $(d=4)$ | **3346.29** (125.3) | 8033.51 (464.0) | 3968.72 (163.9) | | $(\delta=0.5)$ | **3080.56** (119.1) | 5840.21 (373.6) | 6192.34 (373.8) | 3016.91 (133.1) |
| $(d=7)$ | **4405.74** (143.5) | 9675.00 (537.8) | 4107.09 (160.6) | | $(\delta=0.05)$ | **3699.84** (130.1) | 7751.79 (434.2) | 8167.51 (368.3) | 3610.33 (146.1) |
| $(d=10)$ | **5602.55** (180.9) | 9780.54 (360.0) | 4321.84 (167.8) | | $(\delta=0.005)$ | **4297.23** (131.8) | 9810.19 (543.8) | 9278.82 (315.8) | 4219.38 (165.3) |

**Reviewer 1. 1) Experiments.** An important part of our work is the design of the first optimal algorithm whose implementation and performance guarantees are completely independent of the number $K$ of arms. That is why our experiments focused on scenarios with large set of arms. We will include more experiments (essentially all scenarios considered in [16]) – e.g. the table above corresponds to the benchmark used in Soare et al. [12] and all other papers on the topic: here the angle $\omega = 0.1$, the left part of the table is for $\delta = 0.01$, and the right part for $d = 6$. In all experiments we have done, the results suggest that our algorithm is very competitive.

**2) Explore-and-Verify (EV) framework.** Thanks for suggesting that combining RAGE and the EV framework of [13] may lead to an efficient algorithm. As you mention, an idea of the same flavour is used by Chen et al. [0] for combinatorial pure exploration. But to get guarantees in expectation, one needs parallel simulations (Lemma 4.8 in [0]) or repetitive calls of RAGE (see Appendix 2 of [13]). Thus, as already noticed by Fiez et al., the authors of RAGE, the EV framework may be impractical (see [16] Page 7). In fact, neither the authors of [0] nor those of [13] implemented their framework in practice. In our paper, we devise a very simple and asymptotically optimal algorithm using a track-and-stop framework in the spirit of [18]. To this aim, we needed to tackle the following challenges: (i) the optimal allocation is not unique, which poses tracking issues; (ii) computing an optimal allocation in each round might be computationally demanding (we solve this issue with the laziness of our tracking rule); (iii) most of the components of the algorithm, including the stopping rule, must be independent of the number of arms $K$.

**3) Continuous sets of arms.** Our main contribution for such sets of arms is to derive the first (problem-specific) sample complexity lower bound at least for the sphere. This has not been done earlier, and proved to be very challenging. It might be the case that discretizing the set of arms using an $\epsilon/2$-net of the sphere and applying our TS algorithm would yield a sample complexity with the right scaling in $d, \epsilon, \delta$. The algorithm we propose is even simpler, and in particular does not need to work with a set of arms with cardinality growing exponentially with the dimension.

**Reviewer 3. 1) Novelty of the problem and of our solution.** The problem of best arm identification in linear bandits is not new, but it is fundamental. To the best of our knowledge, we propose the first asymptotically optimal algorithm whose implementation and performance guarantees are completely independent of the number $K$ of arms. To this aim and to use the track-and-stop framework [18], we needed to tackle the following challenges: (i) the optimal allocation is not unique, which poses tracking issues; (ii) computing an optimal allocation in each round might be computationally demanding (we solve this issue with the laziness of our tracking rule); (iii) most of the components of the algorithm, including the stopping rule, must be independent of the number of arms $K$. We agree that a very interesting research direction is to derive non-asymptotic performance guarantees for our algorithm.

**Reviewer 4.** Thanks for pointing us to the recent paper of Degenne et al. (it was not available when we submitted our paper). We will cite and discuss it. We also appreciate your suggestions for improving the clarity of the paper. We will elaborate further on the broader impact section as suggested.

**1) Experiments.** Please refer to answer 1) to Reviewer 1.

**2) The constant $c_{\mathcal{A}_0}$.** This constant controls the rate at which forced exploration is performed. Having a too large constant could indeed lead to worse performance. Theoretically, however, it does not affect the asymptotic performance of the algorithm. In practice, we note that forced exploration rarely occurs for the considered level of laziness in our experiments. We conjecture that it may not even be needed. We will provide further discussion on this in the paper.

**3) The goal of $\mathcal{A}_0$.** We confirm that the goal of $\mathcal{A}_0$ is to ensure that the growth rate of $\lambda_{\min}(\sum_{s=1}^t a_s a_s^\top)$ is not too small. We also note that the chosen growth rate $f(t) = O(\sqrt{t})$ is not too large because, ideally, if we knew an optimal allocation, sampling according to it would yield a growth rate of order exactly $t$.

**4) Estimation of $\mu$ along problem-dependent directions.** These directions also appear implicitly in the sampling rule. More precisely, the tracked weights $w_t$ in the sampling rule are solutions of the optimization problem $\max_{w \in \Lambda} \psi(\hat{\mu}_t, w)$.

**5) The stopping rule interpretation.** Intuitively, the stopping rule can be viewed as an empirical version of the problem-specific sample complexity lower bound. Roughly speaking, the stopping rule would correspond to the lower bound where $\mu$ is replaced by $\hat{\mu}_t$, and $w$ by $(N_a(t)/t)_{a \in \mathcal{A}}$.

**6) Continuous set of arms.** Regarding the continuous set of arms, our main result is the lower bound. Deriving such an explicit bound was challenging: in the change-of-measure argument, we needed to propose an appropriate reduction of the set of *confusing* parameters, see Appendix G, Steps 2 and 3 of the proof. We are only able to derive it for the sphere for now. For a generic continuous set of arms, the problem is even harder because in the change-of-measure argument, the aforementioned reduction of the set of confusing parameters becomes even more challenging.

[Meta-Review · NeurIPS 2020]

This paper proposes a "Track-and-Stop" algorithm for linear bandits. Even if the idea is natural, the analysis of such an algorithm requires a few technical challenges, that are well highlighted in the paper and in the rebuttal. Moreover, this paper is the first to suggest and analyze "lazy updates" for Track-and-Stop, which is computationally interesting. We still recommend some clarification in the revision. The rebutall claims that "the implementation and performance guarantees [of the algorithm] are completely independent of K". It is unclear whether this is true for the *implementation* (not the main focus of the paper, though) as computing the oracle w(t) when it needs to be updated probably depends on K. Still regarding the oracle computation, it would be good to explain how it is done precisely. "Frank-Wolfe" is briefly mentioned in the paper, however there is no convergence guarantees for such an algorithm and [Degenne et al. 20] even provide an example in which it does not converge. Some further discussion on this issue would be appreciated. One minor comment to finish: using the acronym "TS" for Track-and-Stop may be misleading, as it often refers to "Thompson Sampling" in the bandit literature.